

# Biogeochemical cycling and phyto- and bacterio-plankton communities in a large and shallow tropical lagoon (Terminos Lagoon, Mexico) under 2009-2010 El Niño Modoki drought conditions

Pascal Conan[1], Mireille Pujo-Pay[1], Marina Agab[1], Laura Calva-Benítez[2], Sandrine Chifflet[3],

Pascal Douillet[3], Claire Dussud[1], Renaud Fichez[3], Christian Grenz[3], Francisco Gutierrez

Mendieta[2], Montserrat Origel-Moreno[2,3], Arturo Rodríguez-Blanco[1], Caroline Sauret[1],

10    Tatiana Severin[1], Marc Tedetti[3], Rocío Torres Alvarado[2], Jean-François Ghiglione[1]

[1] Sorbonne Universités, UPMC Univ Paris 06, CNRS, Laboratoire d'Océanographie

Microbienne (LOMIC), Observatoire Océanologique, F-66650, Banyuls/mer, France

15    [2] Universidad Autonoma Metropolitana, Departamento de Hidrobiología, México D.F.,

Mexico

[3] Aix Marseille Université, CNRS/INSU, Université de Toulon, IRD, Mediterranean Institute

of Oceanography (MIO) UM 110, 13288, Marseille, France

*Correspondence to*: Pascal Conan (pascal.conan@obs-banyuls.fr)





## Abstract

A large set of biogeochemical (nutrients, dissolved and particulate organic matter), phytoplanktonic (biomass and photosynthetic activity) and bacterial (bacterial diversity and ectoenzymatic activities) parameters were determined to understand how the severe drought period relative to the 2009-2010 El Niño Modoki episode influenced biogeochemical cycling and phyto- and bacterio-plankton communities in Terminos Lagoon (Mexico) potentially prefiguring future environmental conditions due to expected trends in climate change. During the study period, the water column of Terminos Lagoon functioned globally as a sink, and especially as a "nitrogen assimilator", because of high production of particulate and dissolved organic matter although exportation of autochthonous matter to the Gulf of Mexico was weak. Coupling between top-down and bottom-up controls accounted for the diverse responses in phytoplankton productivity. Nitrogen and phosphorus stoichiometry mostly accounted for the heterogeneity in phytoplankton and bacteria distribution in the lagoon. In the Eastern part, we found a clear decoupling between areas enriched in dissolved inorganic nitrogen in the North close to Puerto real coastal inlet and areas enriched in phosphate ($PO_4$) in the South close to the Candelaria estuary. Such a decoupling limited the potential for primary production resulting in an accumulation of dissolved organic carbon and nitrogen (DOC and DON, respectively) close to the river mouth. In the Western part of the lagoon, maximal phytoplankton development resulted from the coupling between Palizada river inputs of nitrate ($NO_3$) and particulate organic phosphorus -PP- (but depleted in $PO_4$) and bacterial activity transforming PP and dissolved organic phosphorus (DOP) to available $PO_4$. The Chumpan River only marginally contributed to $PO_4$ inputs due to its very low contribution to overall river inputs. We also found that a complex array of biogeochemical and phytoplanktonic parameters were the driving force behind the geographical distribution of bacterial community structure and activities. Finally, we showed that nutrients brought by the Palizada River supported an abundant bacterial community of polycyclic aromatic hydrocarbon (PAH)-degraders, which are of significance in this important oil production zone.

**Keywords:** biogeochemistry in coastal lagoon, microbial ecology and ecotoxicology, El Niño, lagoon pollution, Gulf of Mexico, Terminos Lagoon



## 1. Introduction

Coastal lagoons are complex environments, combining features of shallow inland water bodies wholly or partly sealed off from the adjacent coastal oceans, influenced by tide, river input, precipitation *versus* evaporation balance and surface heat balance. Interactions between freshwater and marine sources generate strong gradients of salinity, light and nutrient availability (Hauenstein and Ramírez, 1986). Biological diversity is generally high in these environments (Milessi et al., 2010). Located in the Southern Gulf of Mexico near Campeche sound, Terminos Lagoon is one of the largest tropical coastal lagoon worldwide and its recognised environmental importance and protected status is potentially threatened by petroleum-related industrial activities inshore and offshore (García-Ríos et al., 2013). A first tentative budget of salt and nutrient concluded that Terminos Lagoon was slightly autotrophic on a yearly basis (David, 1999), but this assessment was clearly based on scarce environmental data. Chlorophyll-*a* (CHL) concentration and phytoplankton net production have been reported to respectively range from 1 to 17 µg L$^{-1}$ and from 20 to 300 gC m$^{-2}$ a$^{-1}$ (Day et al., 1982), suggesting potential shift from oligotrophic to eutrophic conditions.

In aquatic ecosystems, bacteria utilize a large fraction (up to 90 %) of primary production, since algal carbon exudates might be the principal source for bacterial production (Cole et al., 1988; Conan et al., 1999). Beside the utilization of a considerable part of the available organic matter, bacterioplankton communities also absorb inorganic nutrients thus competing with phytoplankton communities (Conan et al., 2007; Hobbie, 1988). The bulk of organic matter is a highly heterogeneous matrix among which are labile substrates such as proteins or peptides, oligosaccharides, and fatty acids, but most of it is accounted for by complex and refractory substrates (Hoppe et al., 2002). Extracellular enzymes hence are essential to aquatic microorganisms as they allow for the partitioning of complex organic substrates, including high molecular weight compounds which cannot pass through the cell membrane (Arnosti and Steen, 2013). As a function of genetic diversity, the capacity to produce extracellular enzymes is differently distributed in the bacterial community, directly impacting the range of substrates metabolized (Zimmerman et al., 2013). This phenomenon has global-scale implications, since several meta-analysis clearly evidenced differences in the metabolic capacities of microorganisms from temperate, tropical or high latitude waters (Amado et al., 2013; Arnosti et al., 2011). At a local scale, alteration of the evaporation/precipitation balance due to climate change can be challenging especially in the case of coastal lagoon, as it is well known that changes in salinity may alter bacterial diversity and activities (Pedrós-Alió et al., 2000). Local anthropogenic inputs of organic pollutants such as hydrocarbons or herbicides may also be detrimental to bacterial diversity and activities (Aguayo et al., 2014; Rodríguez-Blanco et al., 2010).



Despite their importance, few studies consider the bacterial community in tropical inland aquatic ecosystems (Roland et al., 2010) or in coastal lagoons (Abreu et al., 1992; Hsieh et al., 2012; MacCord et al.,

2013; They et al., 2013) and almost none dealt with tropical coastal lagoons (Scofield et al., 2015). In such a background lacking context, the identified challenge was to foresee how phytoplankton and bacterial abundance, diversity and activity could be affected by incoming climate change in large and shallow tropical lagoons. According to global change scenarios, Centro America is a region where most significant climate change will occur in the second half of the century (Giorgi, 2006) mostly leading to significant decrease in rainfall (Hidalgo

et al., 2013). We therefore took the opportunity of the strong drought conditions related to the 2009-2010 El Niño Modoki episode to sample water in Terminos lagoon, a large and shallow tropical lagoon from the Gulf of Mexico, as a way to assess how expected from long term climate change might affect the structure and functioning of plankton communities. In comparison to the numerous coastal lagoons fringing the Gulf of Mexico, Terminos Lagoon received moderate scientific attention (García-Ríos et al., 2013; Grenz et al., in rev).

Among the existing studies, very few have been conducted on bacterial communities and most of the latter have been based on culture-dependent methods (Lizárraga-Partida et al., 1987; Lizárraga-Partida et al., 1986). However, cultivable bacteria represent a very small fraction of total present bacteria (<0.1 %; Ferguson et al., 1984) and culture-independent methods are requested to more accurately assess the diversity and activity of whole bacterial communities in such a vast and understudied system.

Our study aims at evaluating the links between: i) biogeochemical (nutrients, dissolved and particulate organic matter), ii) phytoplanktonic (biomass and photosynthetic activity) and iii) bacterial (biomass, diversity and ectoenzymatic activities) parameters in the water column of Terminos Lagoon (Mexico) after a sustained period of minimum river discharge relative to the 2009-2010 El Niño Modoki episode. After having identified the main sources of nutrients in the lagoon (focused on nitrogen and phosphorus), we propose a geographical

organization of the ecosystem to explain the distribution of the microbial pelagic communities across the lagoon.



## 2. Materials and methods

### 2.1 Study site and sampling

Terminos Lagoon is a large (1,936 km$^2$, volume 4.65 km$^3$) and shallow (average depth 2.4 m) coastal
lagoon located in the Mexican state of Campeche (Fig. 1), 18°20' to 19°00' N and 91°10' to 92°00' W.
Temperature shows low seasonal variation (27 to 33 ºC), but salinity oscillates from brackish to marine waters
due to high variability in river runoff (Fichez et al., 2016; Gullian-Klanian et al., 2008). River discharge,
precipitation, and groundwater seepage account for 95.44, 4.53 and 0.03 %, respectively, and of the averaged
~12 10$^9$ m$^3$ a$^{-1}$ of freshwater delivered yearly to the lagoon (i.e. about 2.6 times the lagoon volume) the
Chumpan, Candelaria/Mamantel (hereafter Candelaria), and Palizada estuaries account for 5, 19, and 76 %,
respectively (Fichez et al., 2016). The lagoon is connected to the coastal Sea by 2 inlets: El Carmen on the North
Western side (4 km long) and Puerto Real on the North Eastern side (3.3 km). About half of the water volume is
renewed every 9 days, mostly as a result of tidal exchange. The tide is mainly diurnal, with a mean range of
0.3 m (David and Kjerfve, 1998).

Samples were collected at 0.2 m depth at 35 stations distributed over the whole lagoon (Fig. 1) from the
21st to the 27th of October, 2009. In 2009, a yearly cumulated discharge of 4.83 $\pm$ 1.71 10$^9$ m$^3$ broke a historical
deficit record over the 1992-2011 period for the Palizada River (average yearly cumulative discharge of
7.19 $\pm$ 4.22 10$^9$ m$^3$ s$^{-1}$) (Fichez et al., 2016). That exceptional drought period impacted the whole Mesoamerican
region during the 2009-2010 El Niño Modoki episode, and resulted in a salinity positive anomaly in Terminos
Lagoon that most strongly developed during the post wet season period (Fichez et al., 2016), at the time of our
sampling.

A vertical profile of temperature, salinity and fluorescence was carried out at each of the 35 stations with
a SeaBird CTD probe (SBE 19) with a precision of 0.01°C for temperature and 0.001 for salinity. Once the
profile completed, water was sampled using a 5L Niskin bottle maintained horizontally at 0.2 m below the
surface. As soon as the sampling bottle was retrieved on board, a 40 mL Schott® glass vial previously acid
washed was rinsed with sampled water, filled, immediately injected with the fluorometric detection reagent for
ammonia determination, sealed, and stored in the dark for later analysis at the laboratory. Two 30 mL and one
mL plastic acid washed vials were then rinsed with sampled water, filled, stored in a specifically dedicated
and refrigerated ice cooler, to be later deep-frozen at the laboratory awaiting analysis of dissolved inorganic and
organic nutrients. Finally a 4 L plastic acid washed container was rinsed with sampled water, filled, and stored in
a dedicated ice cooler awaiting filtration back at the laboratory.





### 2.2 Nutrients and Dissolved organic matter

Nitrate (NO$_3$ ± 0.02 µM), nitrite (NO$_2$ ± 0.01 µM), phosphate (PO$_4$ ± 0.01 µM) and silicate (Si(OH)$_4$ ±

0.05 µM) concentrations were measured on a continuous flow autoanalyzer Technicon® AutoAnalyzer II

(Aminot and Kérouel, 2007) as previously described in Severin et al. (2014). Ammonium (NH$_4$ ± 10 nM) was

detected at nanomolar concentrations by fluorometric detection (Holmes et al., 1999) on a Turner Design Trilogy

fluorometer.

Samples for dissolved organic matter (DOM) were filtered through 2 precombusted (24h, 450°C) glass

fiber filters (Whatman GF/F, 25 mm). 20 mL were collected for dissolved organic carbon (DOC), into

precombusted glass tubes, acidified with orthophosphoric acid (H$_3$PO$_4$), and analyzed by high temperature

catalytic oxidation (HTCO) (Cauwet, 1999) on a Shimadzu TOCV analyzer. Typical analytical precision is

± 0.1–0.5 (SD) or 0.2–1 % (CV). 20 mL of samples were collected in Teflon vials for dissolved organic nitrogen

(DON) and phosphorus (DOP), and were analyzed by persulfate wet-oxidation according to Pujo-Pay and

Raimbault (1994) and Pujo-Pay et al. (1997).

### 2.3 Particulate organic matter, chlorophyll and phaeopigment

250 mL of seawater were filtered through a precombusted (24h, 450°C) Whatman GF/F glass filters (25

mm), placed into a Teflon vial and oxidized for particulate organic nitrogen (PON) and phosphorus (PP)

measurements (according to Pujo-Pay and Raimbault, 1994). ~1 L was filtered on precombusted (24 h, 450°C)

glass fiber filters (Whatman GF/F, 25mm) for particulate organic carbon (POC) and PON measurements. Filters

were dried in an oven at 50°C and stored, in ashed glass vial and in a dessicator until analyses when return from

the cruise, on a CHN Perkin Elmer 2400.

For chlorophyll (CHL), 250 mL were filtered on 25 mm diameter Whatman® GF/F filters and

immediately stored in liquid nitrogen. CHL and phaeopigment (Phaeo) were later extracted from filters by 100 %

methanol (Marker, 1972) and concentrations were determined by the fluorometric technique (Lorenzen, 1966) on

a Turner Design Trilogy fluorometer.

### 2.4 Photosynthetic parameters

Photosynthetic-irradiance parameters ($\alpha$, P$^b_m$ and I$_k$) were measured using radioactive $^{14}$C-tracer

technique (Fitzwater et al., 1982) in a homemade incubator specifically design. 10x60mL Nunc® culture vials

were cautiously filled and inoculated with Na$_2$H$^{14}$CO$_3$ (final activity of ~0.2 µCi mL$^{-1}$), incubated for 45 min in a

10 light levels irradiance gradient (from 0 to 1327 W m$^{-2}$), then filtered on Whatman GF/F 25 mm filters, rinsed

with 10 % HCl, dried at 45°C for 12 h, and placed into scintillation vials. 10 ml of a liquid scintillation cocktail (Ultima Gold uLLT) were added to the set of scintillation vials 6 h before processing in a Beckman Scintillation Counter. The photosynthetic parameters were determined by fitting each obtained curve with the 'hyperbolic tangent model without photoinhibition' proposed by Jassby and Platt (1976).

### 2.5 Measurements of dissolved total PAH concentrations

Dissolved total polycyclic aromatic hydrocarbons (PAHs) concentrations were determined by using the EnviroFlu-HC submersible UV fluorometer (TriOS Optical Sensors, Germany), a commercially available instrument dedicated to the *in situ* and real time quantification of PAHs in water. The sensor was calibrated in the laboratory before the cruises according to Tedetti et al. (2010) and Sauret et al. (2016). In this work, the mean dissolved total PAH concentrations derived from the sensor are given in ng $L^{-1}$ with a mean coefficient of

variation of 10 %.

### 2.6 Bacterial abundance

Total bacterial counts were determined by flow cytometry (Mével et al., 2008). 2 mL seawater samples were fixed with 2 % formaldehyde for 1 h at 4°C. 1 mL sub-sample was incubated with SYBR Green I (final

conc. 0.05 % [v/v] of the commercial solution) for 15 min at 20°C in the dark and analysed with a FACS Calibur flow cytometer (Becton Dickinson, San Jose, CA) equipped with an air-cooled argon laser (488 nm, 15 mW). Data acquisition and analysis were done with Cell-Quest software (Becton Dickinson).

### 2.7 Total and metabolically active bacterial community structure

Nucleic acids were extracted on 0.2 µm-pore-size filter (47 mm, PC, Nucleopore) by filtration of 1 L of pre-filtered (3 µm) water. Co-extraction of DNA and RNA was performed after chemical cell lysis (Ghiglione et al., 1999) with the Qiagen Allprep DNA/RNA extraction kit using manufacturer instructions. DNA and cDNA (by M-MLV reverse transcription of 16S rRNA, Promega) were used as a template for PCR amplification of the variable V3 region of the 16S rRNA gene (*Escherichia coli* gene positions 329–533; Brosius et al., 1981). The

primer w34 was fluorescently labelled at the 5'-end position with phosphoramidite (TET, Applied Biosystems). CE-SSCP analysis was performed using the 310 Genetic Analyzer and Genescan analysis software (Applied Biosystems), as previously described (Ortega-Retuerta et al., 2012).

### 2.8 Extracellular enzymatic activities



Aminopeptidase, β-glucosidase and lipase were measured using a VICTOR3 spectrofluorometer (Perkin

Elmer) after incubations of 2 h at *in situ* temperature with L-leucine-7amido-4-methyl coumarin (LL, 5 µM

final), MUF-β-D-glucoside (β-Glc, 0.25 µM final) or MUF-palmitate (Lip, 0.25 µM final). These saturated

concentrations and optimized time incubations were determined prior to the extracellular enzymatic activities

measurement, as previously described (Van Wambeke et al., 2009).


### 2.9 Quantification of PAH-degrading bacteria by Most-Probable-Number

The quantification of PAH-degrading bacteria was performed by the most-probable-number (MPN). A

total of 100 µL of each sample was introduced in triplicate in a 48-well microplate with 900 µL of sterile

minimum medium, as previously described (Rodríguez-Blanco et al., 2010; Sauret et al., 2016). A mixture of six

PAHs (naphthalene, fluorene, phenanthrene, fluoranthrene, pyrene and benzo[a]pyrene) diluted in

dichloromethane was introduced into each well at a final concentration of 10 µg mL$^{-1}$. After 2 weeks of

incubation, the change from blue to pink, indicating oxidation of the resazurin contained in the medium was

checked and each sample was analysed by flow cytometry. Classical MPN table gave the most probable number

of bacteria able to degrade the mixture of six PAHs.


### 2.10 Statistical analysis

Comparative analysis of 16S rDNA- or 16S rRNA-based CE-SSCP fingerprints was carried out with the

PRIMER 6 software (PRIMER-E, Ltd., UK) using Bray-Curtis similarities. We used the similarity profile test

SIMPROF (PRIMER 6) to test the null hypothesis of randomly that a specific sub-cluster can be recreated by

permuting the entry ribotypes and samples, when using hierarchical agglomerative clustering. The significant

branch (SIMPROF, p<0.05) was used as a prerequisite for defining bacterial clusters, and clusters were reported

on non-metric multidimensional scaling (MDS) representation.

Canonical correspondence analysis (CCA) was used to investigate the variations in the CE-SSCP

profiles under the constraint of our set of environmental variables, using CANOCCO software (version 5.0), as

previously described in Berjeb et al. (2011). Significant variables (i.e. variables that significantly explained

changes in 16S rDNA- and 16S rRNA-based fingerprintings) in our data set were chosen using a forward-

selection procedure. Explanatory variables were added until further addition of variables failed to contribute

significantly (p< 0.05) to a substantial improvement to the model's explanatory power. Environmental

parameters were previously transformed according to their pairwise distributions, and spearman rank pairwise

correlations between the transformed environmental variables were used to determine their significance.



## 3. Results

### 3.1 Distribution of physical parameters

At the studied period, Terminos Lagoon was characterized by a North West-South East positive gradient of temperature from >30 to about 27°C (Fig. 2A). Salinity was maximal at Puerto Real inlet (37.50) and

along the southern limits of El Carmen Island, intermediate at Candelaria and Chumpan River embouchures, and minimal (21.57) close to the Palizada River (Fig. 2B).

### 3.2 Distribution of biogeochemical parameters

Nitrate and ammonium concentrations (Fig. 2C and 2D) were maximum close to the Palizada

embouchure (16.6 and 0.3 µM, respectively) and to the Puerto Real inlet (2.5 µM in $NO_3$ and the highest $NH_4$ concentration of 1 µM). In the rest of the lagoon, $NO_3$ concentrations were quite low and homogeneous (close to the detection limit of 0.01 µM). $NH_4$ concentrations were more variable with minimum values on the northern side of the lagoon, and concentration in the range 0.1 to 0.3 µM on the southern inshore side.

The distribution pattern for $PO_4$ (Fig. 2E) significantly differed from N-nutrients. Minimum

concentrations (<0.05 µM) were measured in the western part of the lagoon under the influence of the Palizada River, indicating very low $PO_4$ inputs from that river as opposed to nitrogen-nutrients. $PO_4$ concentrations were also low (<0.10 µM) in the centre of the lagoon. The highest $PO_4$ concentration was measured in front of the Chumpan River (0.17 µM), whereas significant inputs in the Eastern part came from Candelaria River (0.13 µM) and Puerto Real inlet (0.12 µM).

The distributions of dissolved organic carbon (DOC; Fig. 2F), nitrogen (DON; Fig. 2G) and phosphorus (DOP; Fig. 2H) concentrations followed a pattern comparable to the one of $PO_4$. The highest concentrations of 400, 20 and 1 µM for DOC, DON and DOP, respectively, were measured in the South Eastern part of the lagoon, either in front of the Chumpan and Candelaria estuaries. Contrary to DOC and DON, DOP concentration was maximal in front of the Chumpan river compare to Candelaria River (82, 95, and 180 % respectively). Lowest

concentrations <200, 5 and 0.1 µM of DOC, DON and DOP were measured in front of the Palizada embouchure and in the case of DOC it even spread along the northern shore of Carmen Island. Significant spearman rank correlations (n=35, p<0.05) were found between DON and DOC (R=0.64), DOP (R=0.64) and temperature (R=-0.32).

The 3 rivers were clearly the main sources of particulate organic nitrogen (PON) and phosphorus (PP)

in the lagoon (Fig. 3). PON reached a maximum concentration of 9.3 µM in front of the Chumpan estuary and progressively decreased while spreading to the North (Fig. 3A). Concerning PP, the Palizada River was the main





source with concentrations close to 0.9 µM, progressively decreasing to 0.6 µM while spreading along the southern shore toward the Chumpan Estuary and 0.5 µM in the north-eastern drift toward Puerto Real passage (Fig. 3B). Significant spearman rank correlations (n=35, p<0.05) were found between PP and PON (R=0.73),

$NO_3$ (R=0.57) and salinity (R=-0.56).

### 3.3 Photosynthetic pigment and activity

Chlorophyll (CHL) and phaeopigment (Phaeo) followed a convergent distribution pattern (Fig. 3C and 3D) with maximum concentrations close or in the vicinity to the Palizada mouth (>6 µgCHL L$^{-1}$ and ~2

µgPhaeo L$^{-1}$). A range of 1-6 µgCHL L$^{-1}$ and 1-2 µgPhaeo L$^{-1}$ was encountered in the western part of the lagoon. Concentrations <1 µgCHL L$^{-1}$ and 1-2 µgPhaeo L$^{-1}$ were mostly confined to the eastern part. On a global view, Phaeo accounted for 28 ± 8 % of CHL on average, hence attesting of rather active phytoplankton communities. Significant spearman rank correlations (n=35, p<0.05) were found between CHL, Phaeo (R=0.82) and POP (R=0.74).

The maximum rate of carbon production per unit of chlorophyll at light saturation ($P^b_m$, Fig. 3E) was minimal (<0.5 mgC mgCHL$^{-1}$ h$^{-1}$) in the Palizada plume in association with the maximum Phaeo:CHL ratio measured (<44 %). Maximum $P^b_m$ values in excess of 8.0 mgC mgCHL$^{-1}$ h$^{-1}$ were measured close the Chumpan estuary in an area of low Phaeo:CHL ratio.

### 3.4 Bacterial abundance and extracellular enzymatic activities

Bacterial biomass (BB) ranged from 1.0 to 4.8 $10^6$ cell mL$^{-1}$ (mean=2.8 $10^6$ cell mL$^{-1}$, SD=0.9 $10^6$ cell mL$^{-1}$, n=35), with maximum values observed in the Puerto Real passage and close to the river embouchures (Candelaria and Chumpan rivers), except for the Palizada river which showed the highest river-lagoon gradient from maximum to minimal values cited above (Fig. 3F).

Cell specific aminopeptidase (Leu-MCA), and phosphatase (MUF-P) activities reached maximum values close to Palizada and Chumpan rivers embouchures (33, and 131.9 fmol L$^{-1}$ h$^{-1}$ cell$^{-1}$, respectively (Fig. 4A, and 4B). Cell specific lipase activity (MUF-Lip) was maximum (10.9 fmol L$^{-1}$ h$^{-1}$ cell$^{-1}$; Fig. 4C) from Chumpan river embouchure northward towards El Carmen Island, crossing the lagoon approximately in its middle following the isotherms (Fig. 2A). Much lower activities were found over most of the lagoon for all the

activities (mean values in fmol L$^{-1}$ h$^{-1}$ cell$^{-1}$ are 12.6 ± 8.4 for Leu-MCA, 12.1 ± 24.2 for MUF-P and 2.4 ± 2.6 for MUF-Lip). Significant spearman rank correlations (n=35, p<0.01) were found between aminopeptidase





activities and DOC (R= -0.27), PON (R=0.33) and between phosphatase activities and $PO_4$ (R= -0.46), PP (R=0.60), $NO_3$ (R=0.69), CHL (R=0.53).

**3.5 Estimated abundance of bacterial PAH-degraders and PAH concentrations**

Quantification by MPN counts showed high enrichment of PAH-degraders close to Palizada river (estimated at $4.6 \ 10^4$ cells $mL^{-1}$, equivalent to 4.4 % of the total bacterial abundance) (Fig. 5A). Lower values were found close to the Chumpan embouchure (estimated at $4.7 \ 10^3$ cells $mL^{-1}$, equivalent to 0.2 % of the total bacterial abundance), and commonly represented less than 0.1 % of the bacterial abundance in the rest of the lagoon. Quantification by MPN counts showed significant even if low spearman rank correlation with dissolved total PAH concentrations (R=0.37, p<0.05, n=35), mainly because of PAH distribution (Fig. 5B) showing higher concentrations close to the El Carmen inlet (332 ng $L^{-1}$) and relatively lower concentrations close to Palizada river (187 ng $L^{-1}$) and to the Chumpan embouchure (166 ng $L^{-1}$). Correlations (p<0.05, n=35) were stronger with PP (R=0.65) and CHL (R=0.53). PAH concentrations were generally lower in the rest of the lagoon (<130 ng $L^{-1}$).

**3.6 Spatial distribution of total and metabolically active bacteria by CE-SSCP fingerprints.**

Bacterial community structure defined as a function of 16S rDNA-based fingerprints from each sample singled out 3 individual stations (Palizada river, El Carmen inlet and Candelaria river) and aggregated 5 groups of stations (Fig. 6A). Three of those groups included a large number of samples: cluster I grouped 9 stations located in the North-eastern part of the lagoon close to Puerto Real inlet; cluster II grouped 9 stations positioned in the middle of the lagoon up North from Chumpan river to the Carmen Island; cluster III grouped 8 stations situated in the South western of the Carmen Island. Two other groups with fewer stations identified intermediated communities found between the El Carmen inlet and the Palizada River in the western part of the lagoon (cluster V; stations 2, 4, 6) and in the middle of the lagoon, close to the Candelaria river (cluster IV; stations 22, 24, 27).

Metabolically active bacterial communities as a function of 16S rRNA-based fingerprints singled out 2 stations (Palizada river and El Carmen inlet) and aggregated 5 groups of stations which are slightly different from the DNA-based clusters (Fig. 6B). Three of those groups included a large number of samples: cluster I formed the largest cluster with 15 stations located in the Eastern part of the lagoon; cluster II grouped 9 stations in the middle of the lagoon up North from Chumpan river to the Carmen Island; cluster III grouped 5 stations in the North Western part of the lagoon, close to El Carmen inlet. Two other groups with fewer stations showed



intermediated communities found close to the Palizada river mouth (cluster IV; stations 6 and 8) and further east (cluster V; stations 9 and 12).

### 3.7 Environmental drivers of the total and active bacterial community structures

To analyse the main environmental factors controlling the spatial distribution of total (Fig. 7A) and active (Fig. 7B) bacterial communities, we performed a canonical correspondence analysis (CCA). In both DNA- and RNA- based analysis, the cumulative percentage of variance of the species-environment relationship indicated that the first and second canonical axis explained 48 % and 24 % of the total variance, respectively for DNA and 45 % and 31 % for RNA. The remaining axes accounted for less than 14 % of the total variance each, and thus were not considered as significant enough.

In the DNA-based CCA, the first canonical axis was positively correlated with $NO_3^-$ and CHL and negatively correlated with concentration of DOC, DOP, DON and oxygen. In the RNA-based CCA, the first canonical axis was positively correlated with $NO_3^-$ and PAHs and negatively correlated with the concentration of POC, PON, oxygen, salinity, $PO_4$ and CHL. The concomitant effect of those parameters explained 27 % and 40 % (ratio between the sum of all canonical eigenvalues and the sum of all eigenvalues) of the changes in bacterial community structure found in the DNA- and RNA-based fractions, respectively (Figure 7).



## 4. Discussion

### 4.1 Biogeochemical characteristics of Terminos Lagoon under low River discharge conditions


With a contribution of about 76 % to river inputs in the lagoon (Fichez et al., 2016; Jensen et al., 1989), Palizada River delivers most of the new nitrogen inputs as nitrate and ammonium. High concentrations in nitrogen were also measured in the Puerto Real inlet, suggesting a second nitrogen source from coastal seawater. These two sources have clearly different impact on primary producer development and activity as shown by the


Phaeo:CHL ratio (<20 % in the vicinity of the Palizada River, but >30 % close to the Puerto Real inlet) and $P^b_m$ values (low in the Palizada area and higher close to the inlet). So, despite greater chlorophyll degradation (indicated by high Phaeo concentrations), phytoplanktonic cells were more productive under the influence of waters from the Gulf of Mexico when compared to those under the river influence. This is in apparent contradiction with what has been classically reported on the influence of rivers inputs in coastal areas that


generally largely enhanced primary productivity (see for example the Rhone River in the Mediterranean Sea; Pujo-Pay et al., 2006). Decreasing turbidity along the estuarine to inlet transect is a first factor explaining the seaward offset of phytoplankton productivity. But higher grazing activity by herbivores in the coastal waters or in the vicinity of the Inlet could further justify the conjunction of higher Phaeo concentrations together with active phytoplankton physiology.. Moreover, Day *et al.* (1982) demonstrated that small additions of filtered


mangrove water had a stimulatory effect on pelagic primary production in Terminos Lagoon. This observation was later confirmed by Rivera-Monroy *et al.* (1998), the latter also evidencing a large temporal variability in stimulating effect, and a rapid inhibition due to variable humic substance concentrations. The relative decrease of productivity close to the Palizada plume could be due to humic matter as we also found relative high concentrations in dissolved PAHs (see hereafter §4.4). Finally, a combined top-down (grazing) and bottom-up


(humic substances) drove the differential responses of phytoplankton productivity in the eastern and western part of the lagoon.

At the time of our study, Palizada River and Puerto Real inlet were major sources of nitrogen to the lagoon. Sediments are generally considered to be a significant internal source of nutrients in shallow coastal ecosystems but they may also be a net sink of dissolved nitrogen at least during certain times of the year


(Sundbäck et al., 2000; Tyler et al., 2003). Rivera-Monroy et al. (1995a) measured nitrogen fluxes between Estero Pargo (an unpolluted tidal creek), and a fringe mangrove forest in Terminos Lagoon. They reported that mangrove sediments were a sink of $NO_3$ and $NH_4$ throughout the year. Denitrification, the dissimilatory reduction of $NO_3$ to produce $N_2O$ and $N_2$, was considered as the main process that contributed to $NO_3$ loss. However, direct measurements of denitrification rates in the fringe and basin mangroves of Terminos Lagoon





indicated low sink of $NO_3$ (Rivera-Monroy et al., 1995b) on the contrary to what has been evidenced in other

mangrove forests (i.e. Twilley, 2013). This was confirmed later by Rivera-Monroy *et al.* (2007) who

hypothesised that most of the inorganic nitrogen was retained in the sediments and not lost via denitrification.

They also measured a decoupling between sources of nitrogen and phosphorus and because P is a limiting

nutrient, they assumed that the dominant source was tidal flooding as opposed to remineralization from organic

matter in the soil. During our study, Origel Moreno (2015) found that benthic carbon mineralization consumed a

large proportion (between 67 and 86 %) of the pelagic carbon production. These values are in the higher part of

the range calculated for sub-tropical lagoons (Grenz et al., 2010; Machado and Knoppers, 1988) and indicate

high biological activity in the sediments. Futhermore, Origel Moreno (2015) estimated that 50 to 95 % of

nitrogen was mineralized in the sediment through various N-consuming processes (see review in Seitzinger,

1988) but also that nitrogen was more efficiently mineralized than phosphorus.

Our large scale study considering the whole lagoon brings some information about the potential origin

of phosphorus in the water column. It is clear from our measurements that phosphate distribution in the lagoon

was disconnected from nitrogen. This impacted the stoichiometry of particulate organic matter (N:P ratio)

through the whole lagoon as shown by the surprising and relative low values of PON:PP ratio (<13) at all

stations (indicating a particulate nitrogen deficit), except for those located in the southwest part of the lagoon

where a canonical Redfied ratio of 16 was measured. To sustain their growth requirement, primary producers

have the ability to decouple their consumption of phosphorus and nitrogen in respect to a variable metabolic

plasticity (Conan et al., 2007). In comparison to the previously discussed 2 main sources of $NO_3$ and $NH_4$

(Palizada River and Puerto Real inlet) located in the West and North part of the lagoon, we identified two

distinct main sources of $PO_4$ in Terminos Lagoon during the sampled period: (i) river inputs from Candelaria and

Chumpan in the South part, even though their contribution to the overall river discharge is low, and (ii)

mineralization of organic phosphorus (PP and DOP) by bacterial activity (coherent with ectoenzymatic activities;

see hereafter §4.2). Note that the major source of PP in the lagoon was the Palizada River, whereas accumulation

of DOP was measured between the Palizada and Chumpan Rivers in the South West of the lagoon. In this area,

distribution of dissolved oxygen was minimal compared to the rest of the lagoon, which was coherent with high

rates of organic matter mineralization in the water column. Finally during our study, the dominant source of $PO_4$

was not tidal flooding as hypothesized by Rivera-Monroy *et al*. (2007), but the mineralization of organic matter

by bacteria in the water column. If that conclusion appears valid in the context of weak river discharges, further

studies will be necessary to test its potential extension to other environmental conditions (rainy periods, river

flood, tidal amplitude...).





### 4.2 Relationship between biogeochemical conditions and bacterial activities

Our analysis of biogeochemical trends in Terminos Lagoon combined with the study of the spatial distribution of prokaryotic extracellular activity. Bacterial aminopeptidase and lipase extracellular activities play

a key function in the transformation of biopolymer into small monomers, since a large part of organic matter is in the form of large size molecules but only small molecules (<600 Da) are directly assimilable by bacteria (Weiss et al., 1991). The expression of aminopeptidase activity indicates the absence of direct bacterial assimilation of dissolved organic matter and their ability to actively release enzymes outside the cells (Van Wambeke et al., 2009). Moderate but significant negative correlations were found between aminopeptidase

activity per cells and DOC concentration in Terminos Lagoon (R=0.27, n=35, p<0.01). The high aminopeptidase activity in the Palizada River plume confirmed the presence of recalcitrant organic matter from terrestrial origin, as opposed to minimum activities in Puerto Real marine waters or in Candelaria embouchures, where DOC concentrations were maximal. Lipase activities showed different trends, with higher activities found in the middle of the lagoon up North from Chumpan River to Carmen Island. We previously published that ambient

quantity and quality of hydrolysable acyl-lipids clearly coupled with the measurement of their *in situ* hydrolysis rates (Bourguet et al., 2009). The differences between spatial distributions of ectoenzymatic aminopeptidase and lipase activities suggest that organic matter from different composition resided in the central zone of Terminos Lagoon, a result in strong agreement with a recent study on hydrodynamics that identified a large circulation cell in the same central area (Contreras Ruiz et al., 2014). Unfortunately, the contribution of the protein or lipid pool

to total organic matter was not measured at the time of the study, which may have strengthen our hypothesis on the role of the composition of organic matter in the spatial distribution of extracellular enzymes activities. This lack of information may explain the very low or absence of correlation found between extracellular activities and measured biogeochemical parameters.

Phosphatase activity is well known to be controlled by the availability of soluble reactive phosphorus

(Van Wambeke et al., 2009). This activity was essentially observed in the vicinity of Palizada River, and not in Puerto real inlet, the two P-depleted zones which indirectly influence the stoichiometry of particulate organic matter, as discussed above. Extracellular phosphatase activity was significantly (p<0.05, n=35) negatively correlated with $PO_4$ (R= -0.46) and positively with PP (R=0.60). Our data therefore converge with the model previously proposed by Robadue *et al*. (2004) predicting a different behaviour between the Eastern and Western

sides of the lagoon in terms of water budget as well as ecosystem functioning, this distinction being mostly





driven by the respective influences of the Palizada River discharge in the West and the Puerto Real marine water inputs in the North East.

### 4.3 Bacterial community structure and ectoenzyme activities

It is generally recognized that the expression of ectoenzyme activities could result from species selection and population dynamics (Martinez et al., 1996) and the zonation of bacterial community structure in the Eastern, middle and Western parts of the lagoon agreed with such a paradigm. The community composition in the Eastern part could be divided into two sub-clusters corresponding to the respective influences of Palizada River mouth and El Carmen inlet. Both DNA- and RNA-based fingerprinting showed that Palizada River and El

Carmen inlet hosted distinct bacterial communities, as previously observed in transition zones such as river (Ortega-Retuerta et al., 2012) or lagoon mouth (Rappé et al., 2000). The relation between community composition and ectoenzyme activities was particularly evident when considering the lipase and aminopeptidase rates. Lipase activity was magnified in the middle of the lagoon with a South to North increasing gradient from Chumpan River to Carmen Island that coincided with specific communities (cluster II in both DNA- and RNA-

based fingerprinting). Other communities were found in the Western part under the influence of the Palizada River, where higher aminopeptidase activity was measured.

        The combination of DNA and RNA strengthen our observations as DNA-based analysis alone would have failed to distinguish between active, dormant, senescent or dead cells hence preventing to assess the level of activity of each detected bacterial population (Rodríguez-Blanco et al., 2010). Even though the abundance of

bacteria in the sea is high, only a small fraction is considered to be metabolically active (Del Giorgio and Bouvier, 2002). Bacterial growth rate has been shown to correlate with cellular rRNA content (Kemp et al., 1993); therefore, information on cellular activity may be obtained by tracking reverse-transcribed 16S rRNA (Lami et al., 2009). The combination of DNA and RNA results in Terminos Lagoon showed similar trends with total and active communities presenting a strong zonation between the eastern, middle and western parts of the

lagoon, to which could be added smaller transition zones located around major sources of coastal (El Carmen inlet) and river inputs (Palizada and Candelaria). These results indicated that most of the communities detected by molecular fingerprinting were active, with no specific distinction through the lagoon.

### 4.4 Biogeochemical parameters and pollutants are driving the bacterial community structure

Molecular fingerprinting (such as CE-SSCP) and next-generation sequencing technologies generally yielded converging results (Ghiglione et al., 2005; Ghiglione and Murray, 2012; Ortega-Retuerta et al., 2012;



Sauret et al., 2015) evidencing clear shifts in bacterial community structure as a function of changes in biogeochemical characteristics (Ghiglione et al., 2005). Numerous factors can regulate microorganism population dynamics, often simultaneously, and several evidences found in the literature (Berdjeb et al., 2011;

Fuhrman et al., 2013; Ghiglione et al., 2008) underlined the importance of relevant statistical analysis to investigate the relative importance of environmental factors in predicting the bacterial community dynamics. Through the use of direct gradient multivariate ordination analyses, we demonstrated that a complex array of biogeochemical parameters was the driving force behind bacterial community structure shifts in Terminos Lagoon. Physico-chemical parameters such as nitrate, oxygen, dissolved organic matter (DOC, DON, DOP) and

chlorophyll *a* acted in synergy to explain bacterial assemblage changes on rDNA level. Some differences were observed to explain the geographical patterns of the metabolically active bacterial communities (rRNA level), where salinity, particulate organic matter (PON, PP) and phosphate were needed in addition to nitrate, oxygen and CHL parameters already outlined on rDNA level. The variance explained by the environmental variables selected by the statistical model only represented 27 % and 40 % of the variability at the DNA and RNA level,

respectively. So, further studies will be needed to elucidate the unexplained variance of the model, due to other parameters not taken into account in our study, such as ecological relationships between bacterial community themselves or top-down control by predation and viral lysis (Ghiglione et al., 2016).

The concentration of dissolved PAHs was also a significant explanatory variable of the metabolically active bacterial community structure. PAHs are considered the most toxic component of crude oil to marine life

and are ubiquitous pollutants in the coastal environment (Kennish, 1991). Our study was performed just before the 2010 Deepwater Horizon (DWH) blowout in the Gulf of Mexico, but several offshore oil platforms exist in the shallow waters of Campeche Bank in the southern part of the Gulf of Mexico, such as the one from the Campeche field (Cheek-1) which is only 60 km North from Terminos Lagoon (Warr et al., 2013). The coast of Campeche itself has been also impacted by the 1979 oil spill of the *Ixtoc I* platform, just about 100 km

Northwest of Terminos Lagoon (Warr et al., 2013). PAHs concentrations in Terminos Lagoon indicated an input into the lagoon from El Carmen inlet (maximal concentration of 332 ng L$^{-1}$) that mostly impacted the Eastern part, with concentration <130 ng L$^{-1}$ in the rest of the lagoon. We observed a high enrichment of PAH-degraders in the South eastern part of the lagoon, with low but significant correlation with PAH concentrations (R=0.37, p<0.05, n=35). This enrichment was particularly high (estimated at 4.6 10$^4$ cell mL$^{-1}$, equivalent to 4.4 % of the

bacterial abundance) in the Palizada River mouth. Nitrogen fertilization from allochthonous inputs from the Palizada River may be crucial for PAHs degradation potential in Terminos Lagoon. Indeed, it is well accepted that bacterial degradation of hydrocarbon (carbon source for bacteria) is dependent on the nutrients to re-



equilibrate the C:N:P ratio (Sauret et al., 2015; Sauret et al., 2016). Some halotolerant bacteria may have the capability to degrade PAHs and survive in river, lagoon and seawater, such as *Marinobacter*

*hydrocarbonoclasticus* sp. 17 (Grimaud et al., 2012). Further studies using PAH-stable isotopes coupled with pyrosequencing (Dombrowski et al., 2016; Sauret et al., 2016) are necessary to identify the dynamic of these functional communities in Terminos Lagoon. Using similar approaches, previous reports showed that the pollutant content and PAHs, in particular, were responsible for the dynamic of bacterial community structure in the sediment of Bizerte lagoon, Tunisia (Ben Said et al., 2010). Such massive impact of pollutants was not

observed here, possibly because of the difference in the degree of pollution between the two areas (moderately contaminated in Terminos Lagoon *versus* highly contaminated in Bizerte). Metabolically active bacterial community structure in the Terminos Lagoon was significantly impacted by PAH-pollution even though it did not exceed the effect of other environmental parameters and their specificity at each geographical location.



### 5. Conclusions


This study provides a new original set of biogeochemical characteristics for one of the largest shallow tropical coastal lagoon and due to the 2009-2010 El Niño Modoki episode, climatic conditions in Terminos Lagoon region were exceptionally dry at the time of our sampling, hence potentially indicative of future environmental conditions resulting from the predicted trends in climate change in the Centro American

region. We evidenced a clear distinction in ecosystem functioning between the east and west parts of the lagoon. Most of the oceanic water entering through the inlets spread toward the south-east where dissolved organic matter accumulated. This area did not support significant phytoplankton development. In the West, a balance shift between a top-down and a bottom-up control explained the different responses in terms of phytoplankton productivity. The decoupling between nitrogen inputs respectively brought by oceanic waters and the Palizada

River, and phosphate inputs from the Chumpan River did not allow for phytoplankton growth. Most of the phytoplankton biomass was aggregated around the Palizada River mouth (which brought most of the freshwater in the lagoon), in a P-depleted area (low phosphate concentration and high bacterial phosphatase activity). Bacterial ectoenzyme activities were mainly observed in the middle of the lagoon, along a south to north cross section stretching from the Chumpan River up to Carmen Island. Maximum mineralization activities were found

in this area, which coincided with high extracellular lipase and aminopeptidase activities and low DOC and $O_2$ concentrations. The lagoon produced significant quantities of particulate and dissolved organic matter thanks to nutrients inputs from the rivers, to uncoupling between nitrogen and phosphate, and to bacterial activities, but in the end most of it was internally processed or stored and only a few of this autochthonous matter was exported to the Gulf of Mexico coastal waters. Hence during our study, the water column of Terminos Lagoon functioned

globally as a sink, and especially as a "nitrogen assimilator". Highest PAH concentrations were measured in El Carmen inlet, suggesting an anthropogenic pollution of the zone probably related to the oil platform exploitation activities in the shallow waters of the South of the Gulf of Mexico and, more locally, to the efflux from El Carmen harbour that serves as a logistical support to the oil extraction industry. We also evidenced the importance of nitrogen fertilization from the Palizada River, which support an abundant bacterial community of

PAH-degraders.

Another significant outcome from our study was (i) to link the spatial distribution of ectoenzymatic activities with changes in bacterial community structure and (ii) to show that a combination of a complex set of physical and biogeochemical parameters was necessary to explain the changes in bacterial community structure. This study also emphasizes the use of direct multivariate statistical analysis to keep the influence of pollutants in





perspective, without denying the role of other physico-chemical variables to explain the dynamic of bacterial community structure in polluted areas.

Our study provided an extensive dataset efficiently mixing biogeochemical status information with information on phytoplankton and bacterial structure and dynamics, which has never been measured before in Terminos Lagoon and its outcomes offers a strong base of information and reflexion for future studies on this

essential coastal system and the potential environmental conditions that might prevail as a consequence of incoming climate change. Further studies will be needed to compare our dataset with high river input regime conditions and asses how it might affect the observed uncoupling between nitrogen and phosphate as well as the dominant source of phosphorus and its consequence on the primary production and bacterial activities.


**Acknowledgments:**

The present work was conducted within the frame of the Joint Environmental Study of Terminos Lagoon (JEST) and jointly financed by the French National Program EC2CO-DRIL, the Institut de Recherche pour le Développement (IRD), the Centre National de la Recherche Scientifique (CNRS), the University of "Lille et des Pays de l'Adour", and the Universidad Autonoma de México (UAM). The authors are strongly gratefully to the

Instituto de Ciencias del Mar y Limnologia, Universidad Nacional Autonoma de México (ICML-UNAM) for providing full access to their field station in Ciudad del Carmen. We also acknowledge P.A. and M.V. Ghighi for carefully proofreading. M.O-M was financially supported by Bonafont S.A De .C.V. and CONACyT during her PhD work.



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



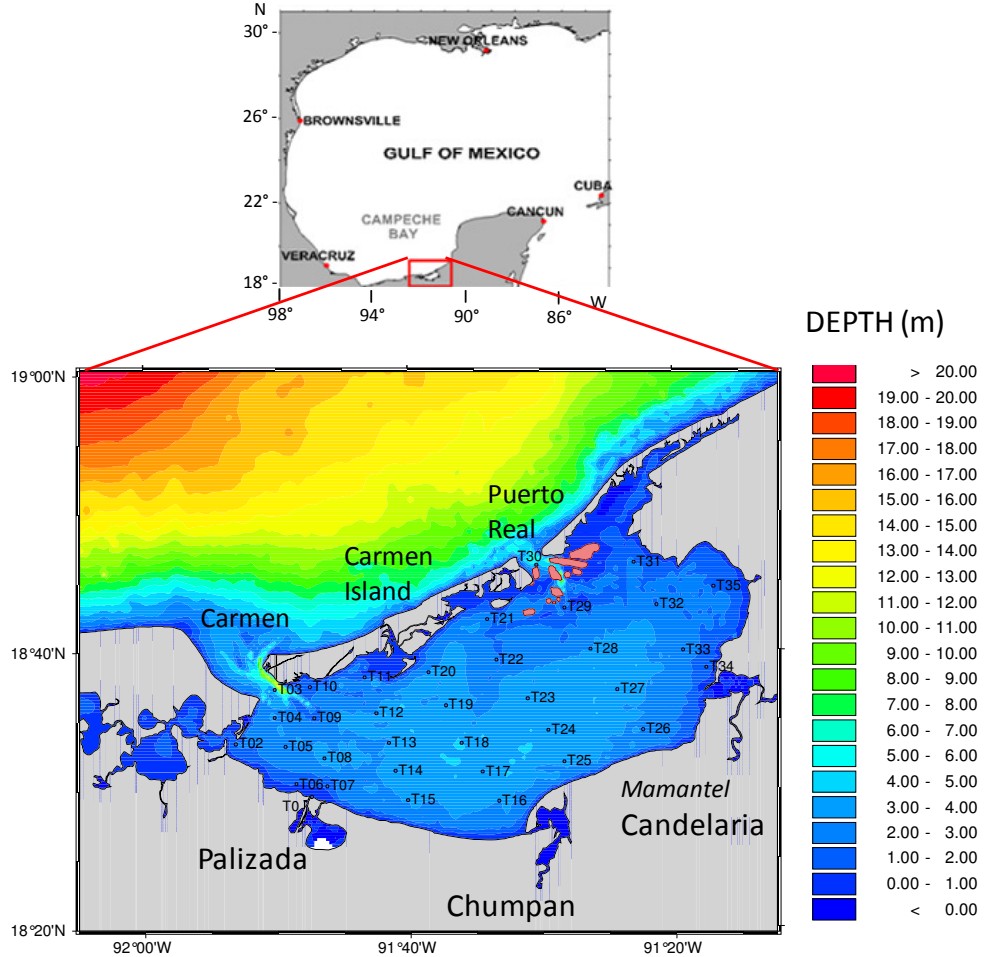

**Figure 1**: Study site location and distribution of the 35 sampled stations in the lagoon.






**Figure 2**: Maps distribution of the physico-chemical parameters measured in the Terminos lagoon in October 2009 for **A.** Temperature (°C); **B.** Salinity; **C.** nitrate concentrations ($NO_3$ in µM); **D.** ammonium concentrations ($NH_4$ in µM); **E.** phosphate concentrations (PO4 in µM); **F.** dissolved organic carbon concentrations (DOC in µM); **G.** dissolved organic nitrogen concentrations (DON in µM); and **H.** dissolved organic phosphorus concentrations (DOP in µM).



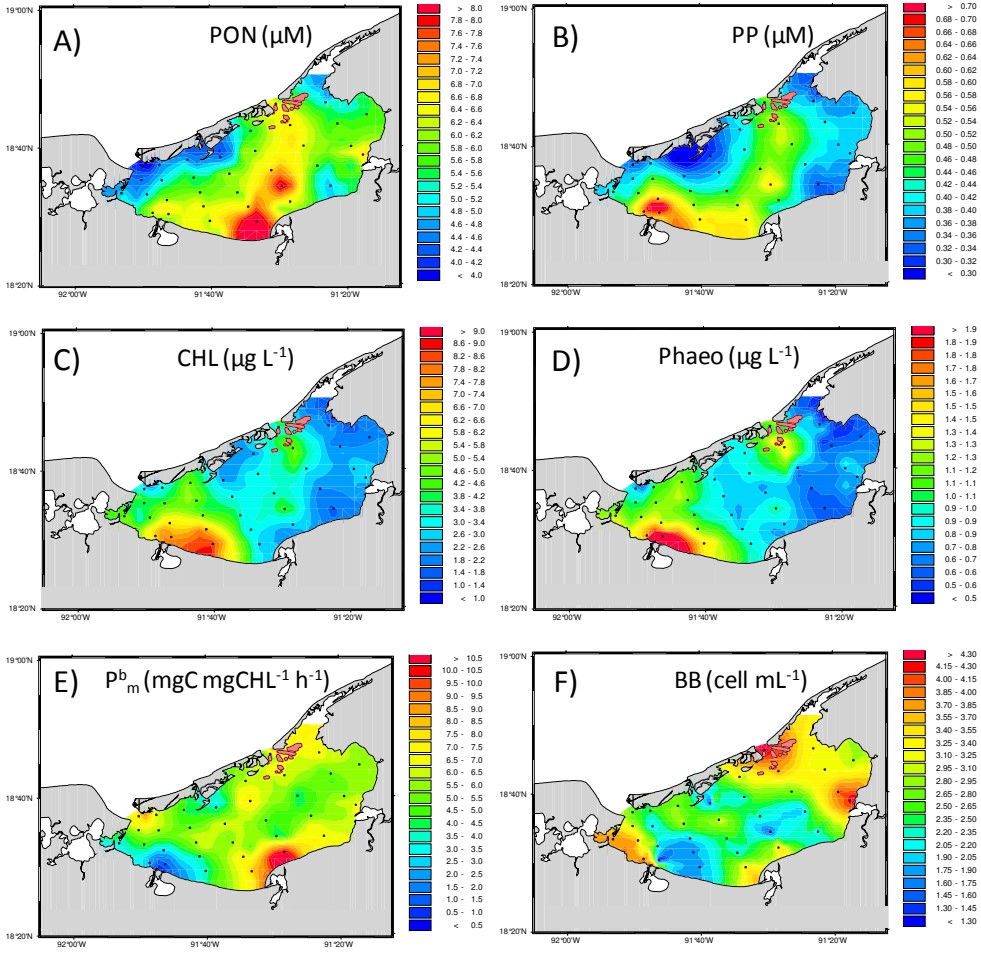

**Figure 3**: as Figure 2 for **A.** particulate organic nitrogen concentrations (PON in μM); **B.** particulate organic

phosphorus concentrations (PP in μM); **C.** Total chlorophyll concentrations (CHL in mg.m$^{-3}$); **D.**

phaeopigments (Phaeo in mg.m$^{-3}$); **E.** maximum photosynthetic rate normalized to chlorophyll (P$^b_m$ in

mgC.mgCHL$^{-1}$.h$^{-1}$); **F.** bacterial abundance ($10^6$ cell.mL$^{-1}$)





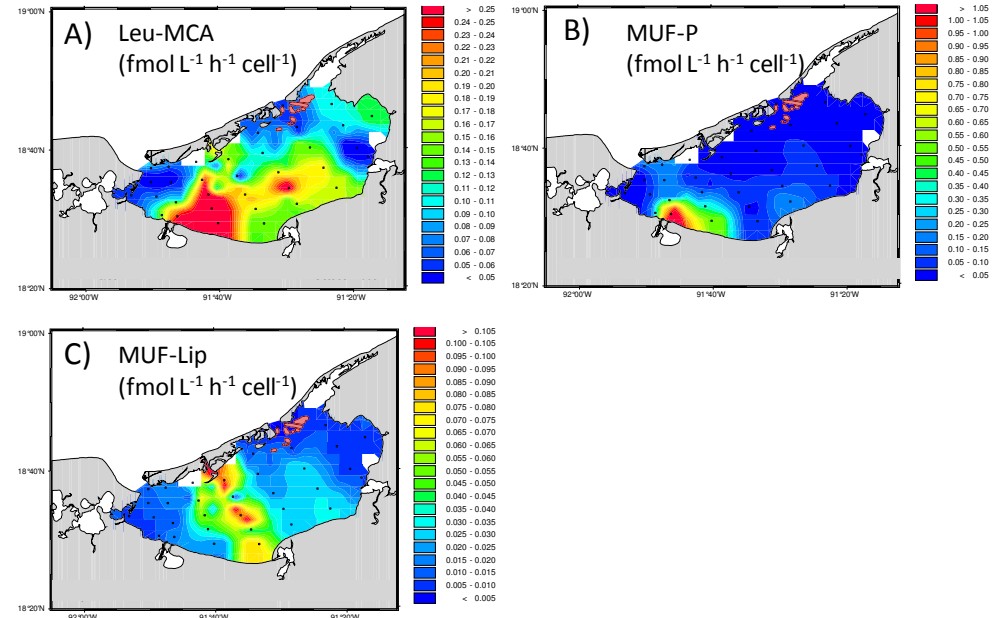


**Figure 4**: as Figure 2 for **A.** aminopeptidase activities (fmol.L$^{-1}$.h$^{-1}$.cell$^{-1}$); **B.** phosphatase activities (fmol.L$^{-1}$.h$^{-1}$.cell$^{-1}$); and **C.** Lipase activities (fmol.L$^{-1}$.h$^{-1}$.cell$^{-1}$)




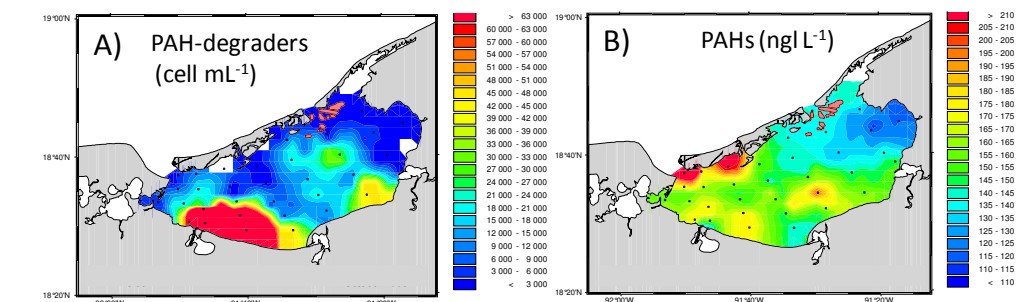


**Figure 5**: as Figure 2 for **A.** the most-probable-number (MPN in count); and B. Total dissolved PAHs (ng.L$^{-1}$)



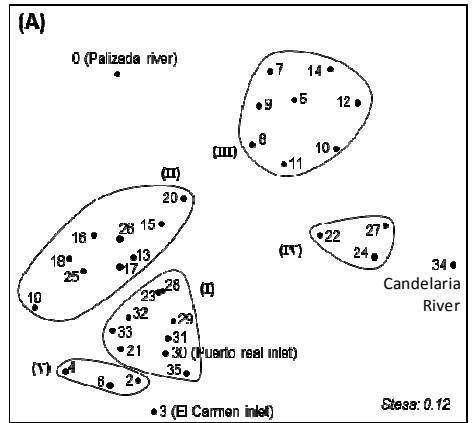
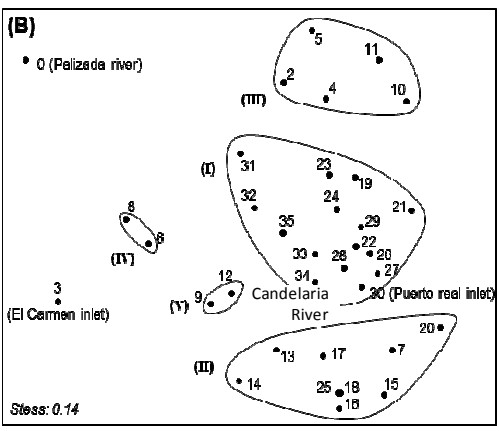


**Figure 6:** Multidimensional scaling (MDS) plot of the total (**A**) and metabolically active (**B**) bacterial community structures as determined from CE-SSCP profiles based on Bray–Curtis similarity index. Clusters were determined according to similarity profile test SIMPROF (p<0.05).





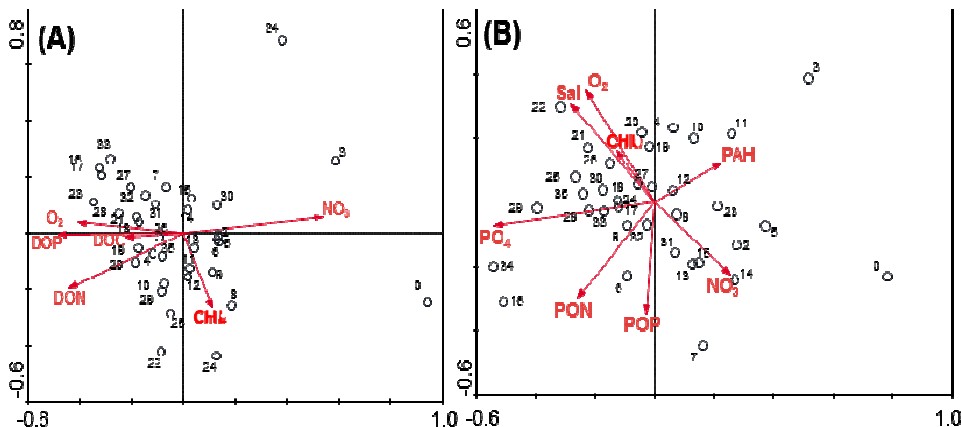


**Figure 7:** Canonical correspondence analysis of total (A) and active (B) bacterioplankton community structure

from the 35 samples using physico-chemical parameters. Arrows point in the direction of increasing values

of each variable. The length of the arrows indicates the degree of correlation with the represented axes. The

position of samples relative to arrows is interpreted by projecting the points on the arrow and indicates the

extent to which a sample bacterial community composition is influenced by the environmental parameter

represented by that arrow. The variance explained by the environmental variables selected by the model

represent 27 % and 40 % of the variability at the DNA and RNA level, respectively.