# Peer review of "Biogeochemical cycling and phyto- and bacterio-plankton communities in a large and shallow tropical lagoon (Terminos Lagoon, Mexico) under 2009-2010 El Niño Modoki drought conditions"

_Biogeosciences, 2016_

## Referee Comment (RC1) · Anonymous Referee #1 · 1 Sep 2016

GENERAL COMMENT : Overall an interesting case study. Our understanding of such coastal ecosystem, at the interface between the continental and the marine environments in a context of anthropization and global change depends on such studies. The main force of this work is the link between the biological and geochemical points of view. However, such link could be reinforced by improving the discussion section. Moreover, several more or less severe issues need to be addressed all along the manuscript before publication. No additional data is needed, but complementary data anlysis could easily promote the quality of the work.

SPECIFIC COMMENTS

Abstract: - The top-down control is not demonstrated in this work, you cannot mention this so securely as a major driver, neither as a result of your work. Same remark for the discussion section.

Introduction: - The part related to climate change is really oversized when compared to real scientific questions addressed by this paper in its current version. Moreover, the consideration of PAH rather balance the study towards local changes because of human activities. This introduction should at least be less oriented towards climate change and present the challenge of understanding the functioning of such continental to marine environments interface in a context of growing anthropization, which seems to fit better to the design and results of the study.

- The consideration of PAH is not mentioned in the introduction. It is surprising to see it appear in the methods section without any mention in the objectives! The results and the discussion related to PAH are interesting, but you need to include it in your objectives, otherwise it is really hard to see how useful it can be to consider PAH regarding to your scientific objectives. This comment and the previous one are really connected, they should be taken into consideration together. The way PAH are mentioned in the abstract, the importance of oil-related compounds is obvious, this should be presented in this way in the introduction.

Material and methods:

- Section 2.6 and later all along the manuscript: flow cytometry cannot distinguish bacteria from archaea, thus most of the "bacterial" should be replaced by "prokaryotic" or "heterotrophic prokaryotes".

- Section 2.9: Could you explicit the calculation of this MPN? The way such calculation is performed after two weeks of incubation sounds weird, how is the number of bacteria enumerated after two weeks related to the initial sample? Also refer to my

corresponding comment in the result section.

Results: - Lines 279-280: very consistent, but could you indicate the approximate value of the ratio in this area?

- Lines 292-294: the correlations concerning aminopeptidase activities are very weak, it should not be presented at the same level than the ones concerning phosphatase activities, which are stronger. This probably means that there is indeed a correlation between these variables, but they are not linear and Spearman's correlation tests the linearity of the relation. You should try non parametric and non linear correlation analyses to further precise the link between these variables.

- Lines 296-299: Need to explain how such percentage is obtained. As it is explained in the current version of the manuscript, one could understand that you counted bacteria in your 2 week enrichments and divided this number by the number of bacteria in your initial samples... This does not sound correct.

- Lines 300-303: I don't understand the cause-to-consequence relationship you mention here (second part of the sentence). You should split the sentence into two very descriptive ones. Moreover, the PAH distribution should be presented at the beginning of this paragraph, then followed by PNM counts, and finally the correlation analysis. Thus the two components of the title should also be switched.

Discussion: - Lines 346-348: The corresponding figures are puzzling because, yes, the normalized productivity per cell seems to be higher, but since the CHL concentration is also lower, one could think that the productivity of the area expressed by unit of volume could be lower. Since you have CHL concentration per liter and since the productivity is expressed is expressed by unit of CHL, I suggest that you calculate a productivity by volume of water in order to better compare the productivity of the two sites. According to your graph, you have less than a 2-fold difference in CHL concentration between the two sites but a 6- to 7-fold difference in C fixation rates. Thanks to such analysis, your conclusion will be more robust.

- Lines 352-353: Could you provide references to support such hypothesis since you did not measure it?

- Lines 359-361: This sentence should be rewritten in a more prudent way since it is only speculation, you have no clue for the top down control and even though you have PAHs concentrations, the direct link with phytoplankton productivity remains to be demonstrated.

- Lines 378-380: sounds weird to justify a recent study by an old one... The old reference could be removed without making the sentence meaningless or doubtful.

- Lines 383-386: These Redfield ratios were not presented in the results sections. They are meaningful and should be presented extensively!

- Lines 410-413: That could appear contradictive. If you suppose that higher concentrations of DOC but lower aminopeptidase activity suggest a higher amount of labile organic matter for bacteria, you should clearly state it.

- Lines 425-427: So vincinity of Palizada river = phosphatase activity but P-depleted zone, meaning very low P-availability for phytoplanktonic growth, which seems consistent with le smaller C fixation rates observed in this zone, am I right? If yes, such a link between nutrients and biological productivity could be added to this discussion, this would greatly strengthen the end of the first sub-section which was up to now very speculative. Moreover, this would also strengthen the links between the geochemical and the biological sides of this paper which appear very few connected for the moment.

- End of section 4.3: A small discussion could be added about distance-based effects and community turnover: according to your data, one could think that strong local selective pressures led to strong shift in the community composition, instead of having progressive gradients and/or fast dispersion that would rather just promote a turnover within the active fraction. Such thinking points out the existence of strong driving pressures that you did not measured in your study, since you do not have such strong

correlations.

- Lines 460-466: These two sentences should rather be placed at the beginning of the previous paragraph, before discussing about community structure.

- Lines 475-477: Adding a short discussion about "distance-based" similarities in present and active comunities, as proposed in a previous comment would perfectly fit with this discussion.

- The link between the severe drought and the potential predictive aspect of this study is not mentioned all along the discussion section, whereas it represents most of your introduction. You need either to remove this "climate-change" part in the intro, or discuss it in the discussion.

Conclusion - Lines 512-514: You cannot say that as it was unambiguous, you did not even measure any top-down parameter.

- Lines 514-517: This sentence does not sound correct since you measured C fixation, thus phytoplankton activity. To measure groth, you need repeated measures in time, that you do not have. So you can definitely not state that there is no growth. I understand it is probably not what you wanted to say, but your sentence could be misinterpreted

- Lines 528-530: again, you are over confident in your hypothesis: "seems to support", you do not have any temporal data to support this hypothesis.

TECHNICAL CORRECTIONS

L 101: replace bacterial by prokaryotic, except if the enzymatic activities have been demonstrated to be specific to bacteria and not to archaea...?

L 112-116: split this long sentence into two, after the first "respectively".

L 185: rather indicate the final concentration like "0.5X" and precise the name of the company which provided the solution

L 187: provide the optical parameters used to discriminate heterotrophic prokaryotes from the rest (side scatter and green fluorescence?)

Section 2.10: the software used to perform the correlation analyses is not précised.

L 253: "... either in front of the Chumpan river OR..."

L 253-254: Rephrase: "Maximal DOC and DON concentrations (82% and 95%, respectively) were measured in front of Candelaria river, whereas the maximal DOP concentrations was observed in front of the Chumpan river"

L 256-257: here and latter on: "Spearman's rank correlation", and use the lowercase greek letter "rho" instead of R.

L 273-274: rephrase: "...were found between CHL and Phae (R=0.82) or PP (R=0.74)". And use PP instead of POP (which was not defined but I assume it means the same as PP).

L 277: "...(<44%)...": since it is a maximal value that you mention here, it would be more logical and precise to indicate "superior to..." (>) instead of <44%, because in agreement to your sentence this ratio is always <44%.

L 303-304: correlation between PP or CHL and which other variable? MPN counts? need to specify it

L 323: "intermediate" (no "d")

L 356: change "the latter also evidencing..." in "who also evidenced..."

L 358: "relativeLY"

L 472: "where" => which

---

## Referee Comment (RC2) · Anonymous Referee #2 · 17 Sep 2016

GENERAL COMMENT:

Conan et al propose in this paper an interesting case study providing records of biological and geochemical variables in a large and shallow tropical lagoon during an important drought period. This study gives an original overview of the potential links between biological and geochemical variables taking place in this environment during such climatic condition. Such data are still scarce in the literature and are urgently needed to better understand the dynamic of these transitional land/sea areas under direct and indirect pressures. However, these data were obtained during a single sampling campaign without any replication. The authors need to be careful about the importance of this study as a way to assess how long-term climate change may affect the structure and functioning of planktonic communities (as mentioned in the Introduction). This study is a snapshot of the situation at a given time that cannot be extended to the understanding of climate change effects on lagoon ecosystems. The main force of this study is the link between geochemical variables and microbial communities structure/function. Globally, this manuscript is well organized, excepted for introduction and objectives sections that don't mention anywhere why PAH and PAH-degraders were specifically analyzed further. The demonstration of the link between PAH, enzymatic activities and microbial community diversity is an important section of the study and introduction need to be revised in order to better reflect the results discussed by the authors. This study produces quite interesting original results but, in its present form, the aims presented in the introduction do not fit well with the results and discussion presented, especially concerning the PAH-relative measurements.

SPECIFIC COMMENTS

Abstract:

- The abstract states that the study will help to "understand how the severe drought period …. influenced biogeochemical cycling and phyto- and bacterio-plankton communities", nevertheless the study does not compare the data obtained during this period to any other records (maybe due to the fact that they do not exist) obtained during in more "classic" situation. This statement needs to be nuanced.

- "Coupling between top-down and bottom-up controls accounted for the diverse responses in phytoplankton productivity" : I didn't see any demonstration of top-down control in the study.

- The PAH-relative measurements need to be better integrated in the global aim of the study.

Introduction:

- Overall the introduction is too much climate change orientated in regards to the results presented in the manuscript. Introduction needs to be more focused on the local dynamic of this lagoon and on the importance of considering PAH in this specific area. This study is more focused on the dynamic of land/sea transitional coastal areas under combined climatic and chemical pressures than on climate change effects.

- The importance of PAH in the studying area is not mentioned anywhere in the introduction or in the objectives. However, PAH concentrations and PAH-degraders appeared as a substantial part of the results and discussion sections. If PAH is an important factor to consider in the lagoon, this should be clearly presented in the objectives of the study.

Material and Methods:

- Section 2.1 (last paragraph): I suggest including the sub-sampling information in the relative subsequent technical sections Is there any available data concerning water circulation in this lagoon ? This information could be relevant for your study.

- Section 2.6: Only free bacteria can be measured according to this protocol not "total bacteria" as mentioned in the text (attached bacteria are not measured)

- Section 2.7: In turbid waters, such prefiltration step may lead to the retention of attached bacteria on the filter and induce a bias in bacterial diversity assessment. This method is adequate to collect the bacteria present in aquatic sample but this bias need to be mentioned in the discussion section.

- Section 2.9: The authors used a mixture of 6 PAHs for MPN but there is no explanation of this choice. In addition, what is the ratio of each PAH in the mixture? Why did they use a 10 $\mu$g/mL final concentration (is this concentration realistic when dealing with environmental samples)? Additional information is required in this section.

Results:

- PAH concentrations should be included in section 3.2

- Section 3.4: Biomass is the total amount of living material in a given habitat, population, or sample. Specific measures of biomass are expressed in dry weight per unit area of land or unit volume of water. In this section, you report bacterial abundance (cells/ml) and not bacterial biomass.

- The standard deviation values for MUF-P and MUF-Lip are higher than the measured mean values. The use of a range of values would be more suitable. In addition the values presented in the text do not fit with the values presented in the legend of Figure 4. This needs to be corrected.

- Section 3.5: As previously mentioned, flow cytometry does not consider attached bacteria. The protocol used for MPN determination allows the growth of free and attached PAH-degraders. As a consequence, the measured percentage is not accurate.

Discussion

- Lines 352-361: No grazing activity was measured in this study, so this statement need to be presented as a other way to explain the data and not as an affirmation.

- Section 4.3 : Please explain the last sentence of the section

Conclusion

- No demonstration of any top-down control in this study

- The conclusion gives a nice overview of the main results obtained in the lagoon but does not respond to the climate change questions mentioned in the introduction. This needs clarification or rephrasing of the introduction.

TECHNICAL AND TYPOS CORRECTIONS

L59: environmental importance and protected status are . . .

L81: Are hydrocarbons always detrimental to bacterial diversity and activity?

L131: what is the fluorometric detection reagent used? How long were the samples stored before being processed in the laboratory

L183: Indicate the name of the company providing SYBR Green

L282: river mouths

L285-L290: values do not fit with the values presented in Figure 4

L403: Verify the first sentence of section 4.2

L784 : Put the 4 in index for phosphate

Figure 5B : Correct the unit of PAHs concentration on the figure

---

## Author Comment (AC1) · 22 Nov 2016

SPECIFIC COMMENTS Abstract: - The top-down control is not demonstrated in this work, you cannot mention this so securely as a major driver, neither as a result of your work. Same remark for the discussion section.

Answer. Yes indeed, so we replaced this sentence by "We found that "bottom-up" control accounted for a large part in the variability of phytoplankton productivity". We also modified the discussion in this way and only suggested the top-down as a potential factor.

Introduction: - The part related to climate change is really oversized when compared to real scientific questions addressed by this paper in its current version. Moreover, the consideration of PAH rather balance the study towards local changes because of human activities. This introduction should at least be less oriented towards climate change and present the challenge of understanding the functioning of such continental to marine environments interface in a context of growing anthropization, which seems to fit better to the design and results of the study.

Answer. We deleted the whole paragraph referenced to global change and added the challenge put forward by this remark.

- The consideration of PAH is not mentioned in the introduction. It is surprising to see it appear in the methods section without any mention in the objectives! The results and the discussion related to PAH are interesting, but you need to include it in your objectives, otherwise it is really hard to see how useful it can be to consider PAH regarding to your scientific objectives. This comment and the previous one are really connected, they should be taken into consideration together. The way PAH are mentioned in the abstract, the importance of oil-related compounds is obvious, this should be presented in this way in the introduction.

Answer. We agree with the Reviewer. We thus added two paragraphs dedicated to PAHs in the introduction: "Local anthropogenic inputs of organic pollutants such as polycyclic aromatic hydrocarbons (PAHs) may also affect bacterial diversity and activities (Lekunberri et al., 2010; Rodríguez-Blanco et al., 2010; Jiménez et al., 2011). Indeed, PAHs, which can comprise as much as 25–35 % of total hydrocarbon content in crude oils (Head et al., 2006), are among the most abundant and ubiquitous pollutants in the coastal environment (González-Gaya et al., 2016). These compounds are recognized by the European and US environmental agencies as priority pollutants for the aquatic medium due to their toxicity, persistence and ability to accumulate in the biota (Kennish, 1992). Hence, the presence of PAHs in the marine environment may induce an increase in the indigenous populations of marine bacteria that can break

down and utilize these chemicals as carbon source, the so called "PAH-degrading bacteria" or "PAH degraders". These bacteria are generally strongly selected in oil-impacted ecosystems, where they may account for 70 to 90% of the total bacterial community (Head et al., 2006; Gutierrez et al., 2014)." "Also, Terminos Lagoon is potentially impacted by PAHs, which may come from a diversity of sources including sea-based activities (spills from ships, platforms and pipelines, ballast water discharge, drilling. . .) but also rivers, surface runoffs and the atmosphere that carry various urban and industrial wastes (fuel combustion, traffic exhaust emissions...). Nevertheless, to our knowledge, little is known about the PAH content in this ecosystem. Even though Noreña-Barroso et al. (1999) have reported PAH concentrations in the American Oysters Crassostrea virginica and Rendon-Von Osten et al. (2007), PAH concentrations in surface sediments, no data are currently available about dissolved PAH concentrations in surface waters of the Terminos Lagoon."

In the objectives, we also added "dissolved PAHs" and "including PAH-degrading bacteria".

We removed one reference from the reference list: Aguayo, P., Gonzalez, C., Barra, R., Becerra, J., and Martinez, M.: Herbicides induce change in metabolic and genetic diversity of bacterial community from a cold oligotrophic lake, World journal of microbiology & biotechnology, 30, 1101-1110, 2014. And we added several references dealing with PAHs/PAH-degrading bacteria:

González-Gaya, B., Fernández-Pinos, M. C., Morales, L., Abad, E., Piña, B., Méjanelle, L., Duarte, C. M., Jiménez, B., and Dachs, J.: High atmosphere-ocean Exchange of semivolatile aromatic hydrocarbons, Nature Geoscience, 9, 438-442, 2016. Gutierrez, T., Rhodes, G., Mishamandani, S., Berry, D., Whitman, W. B., Nichols, P. D., Semple, K. T., and Aitken, M. D.: Polycyclic Aromatic Hydrocarbon Degradation of Phytoplankton-Associated Arenibacter spp. and Description of Arenibacter algicola sp. nov., an Aromatic Hydrocarbon-Degrading Bacterium, Applied and Environmental Microbiology, 80, 618-628, 2014. Head, I. M, Martin Jones, D., and Roling, W. F. M.: Marine microorganisms make a meal of oil, Nature, 4, 173-182, 2006. Jiménez, N., Viñas, M., Guiu-Aragonés, C., Bayona, J. M., Albaigés, J., and Solanas, A. M.: Polyphasic approach for assessing changes in an autochthonous marine bacterial community in the presence of Prestige fuel oil and its biodegradation potential, Applied Microbiolal Biotechnology, 91, 823-834, 2011. Lekunberri, I., Calvo-Díaz, A., Terira, E., Morán, X. A. G., Peters, F., Nieto-Cid, M., Espinoza-González, O., Teixeira, I. G., and Gasol, J.M.: Changes in bacterial activity and community composition caused by exposure to a simulated oil spill in microcosm and mesocosm experiments, Aquatic Microbial Ecology, 59, 169-183, 2010. Noreña-Barroso, E., Gold-Bouchot, G., Zapata-Perez, O., and Sericano, J. L.: Polynuclear Aromatic Hydrocarbons in American Oysters Crassostrea virginica from the Termines Lagoon, Campeche, Mexico, Marine Pollution Bulletin, 38, 637-645, 1999. Rendon-von Osten, J., Memije, M., Ortiz, A., and Benitez, J.: Potential sources of PAHs in sediments from Terminos lagoon, Campeche, Mexico, Toxicology Letters, 172, S162, 2007.

Material and methods: - Section 2.6 and later all along the manuscript: flow cytometry cannot distinguish bacteria from archaea, thus most of the "bacterial" should be replaced by "prokaryotic" or "heterotrophic prokaryotes".

Answer. We have done the replacement in the whole text when appropriated.

- Section 2.9: Could you explicit the calculation of this MPN? The way such calculation is performed after two weeks of incubation sounds weird, how is the number of bacteria enumerated after two weeks related to the initial sample? Also refer to my corresponding comment in the result section.

Answer. Sorry, but the reviewer's remark is not very clear. The principle of the MPN technique is based on serial dilution on the reference sample to be analysed. Therefore, the calculation takes into account the initial number of bacteria and the dilution where resazurin changed color. It makes no sense to count the bacteria after two weeks incubation and use this count in the calculation. Here, the MPN is giving an estimation of the most probable number of bacteria able to degrade the mix of six PAHs as sole carbon source within two weeks. We reassure the reviewer that the use of classical MPN table based on symmetrical dilution series with a constant, table-specified dilution factor, as explained in the material and method section.

A reference has been added in the material and method section, in case the reader wants to have more information on this method classically used in microbiology. Alexander, M., 1982. Most probable number method for microbial populations. In: Page, A.L., Miller, R.H., Keeney, D.R. (Eds.), Methods of Soil Analysis, Part 2, 2nd ed. American Society of Agronomy, Madison, WI, pp. 815-820.

Results: - Lines 279-280: very consistent, but could you indicate the approximate value of the ratio in this area?

Answer. We indicated in the text the Phaeo:CHL ratio of the area (<25%)

- Lines 292-294: the correlations concerning aminopeptidase activities are very weak, it should not be presented at the same level than the ones concerning phosphatase activities, which are stronger. This probably means that there is indeed a correlation between these variables, but they are not linear and Spearman's correlation tests the linearity of the relation. You should try non parametric and non linear correlation analyses to further precise the link between these variables.

Answer. We now precised ". . .. to a lesser extent, between phosphatase activities. . .."

- Lines 296-299: Need to explain how such percentage is obtained. As it is explained in ,the current version of the manuscript, one could understand that you counted bacteria in your 2 week enrichments and divided this number by the number of bacteria in your initial samples... This does not sound correct.

Answer. See our previous response on - Section 2.9 for this remark.

- Lines 300-303: I don't understand the cause-to-consequence relationship you mention here (second part of the sentence). You should split the sentence into two very

descriptive ones. Moreover, the PAH distribution should be presented at the beginning of this paragraph, then followed by PNM counts, and finally the correlation analysis. Thus the two components of the title should also be switched.

Answer. We agree with the reviewer. We modified this paragraph 3.5 accordingly: We replaced the paragraph title "3.5 Estimated abundance of bacterial PAH-degraders and PAH concentrations" by "3.5 Dissolved PAH concentrations and estimated abundance of bacterial PAH-degraders"

And we replaced the whole concerned paragraph: "Quantification by MPN counts showed high enrichment of PAH-degraders close to Palizada river (estimated at 4.6 104 cells mL-1, equivalent to 4.4 % of the total bacterial abundance) (Fig. 5A). Lower values were found close to the Chumpan embouchure (estimated at 4.7 103 cells mL-1, equivalent to 0.2 % of the total bacterial abundance), and commonly represented less than 0.1 % of the bacterial abundance in the rest of the lagoon. Quantification by MPN counts showed significant even if low spearman rank correlation with dissolved total PAH concentrations (R=0.37, p<0.05, n=35), mainly because of PAH distribution (Fig. 5B) showing higher concentrations close to the El Carmen inlet (332 ng L-1) and relatively lower concentrations close to Palizada river (187 ng L-1) and to the Chumpan embouchure (166 ng L-1). Correlations (p<0.05, n=35) were stronger with PP (R=0.65) and CHL (R=0.53). PAH concentrations were generally lower in the rest of the lagoon (<130 ng L-1).

By the paragraph: "Dissolved total PAH concentrations (Fig. 5A) were higher close to the El Carmen inlet (332 ng L-1) and relatively lower close to Palizada river (187 ng L-1) and to the Chumpan embouchure (166 ng L-1). They were generally lower in the rest of the lagoon (<130 ng L-1). Quantification by MPN counts showed high enrichment of PAH-degraders close to Palizada river (estimated at 4.6 104 cells mL-1, equivalent to 4.4 % of the total bacterial abundance) (Fig. 5B). Lower values were found close to the Chumpan mouth (estimated at 4.7 103 cells mL-1, equivalent to 0.2 % of the total bacterial abundance), and commonly represented less than 0.1 % of the bacterial

abundance in the rest of the lagoon. Quantification by MPN counts showed significant even if low spearman rank correlation with dissolved total PAH concentrations (R=0.37, p<0.05, n=35). Correlations (p<0.05, n=35) were stronger with PP (R=0.65) and CHL (R=0.53)."

Consequently, we inverted the order of Fig. 5A and 5B: Figure 5: as Figure 2 for A. total dissolved PAHs (ng L-1) and B. the most-probable-number (MPN in count)

Discussion: - Lines 346-348: The corresponding figures are puzzling because, yes, the normalized productivity per cell seems to be higher, but since the CHL concentration is also lower, one could think that the productivity of the area expressed by unit of volume could be lower. Since you have CHL concentration per liter and since the productivity is expressed is expressed by unit of CHL, I suggest that you calculate a productivity by volume of water in order to better compare the productivity of the two sites. According to your graph, you have less than a 2-fold difference in CHL concentration between the two sites but a 6- to 7-fold difference in C fixation rates. Thanks to such analysis, your conclusion will be more robust.

Answer. Thanks for this remark. In fact, we have actually about 2-fold difference in CHL concentration between the two sites but about the same ratio and not 6- to 7-fold difference in C fixation rates. So, for similar amount of nutrients (and DOM and DOP also. . .) we have similar production for half chlorophyll. . . Here is the table for mean values of the 2 groups (I included 4 stations in each group. . .) I also added a sentence in the text concerning this comparison.

CHL Pbm PP max NO3 PO4 NH4 mg.m-3 mgC.mgCHL-1.h-1 mgC.m-3.h-1 $\mu$M $\mu$M $\mu$M mean G palizada 6.34 4.34 27.5 0.02 0.08 0.05 mean G inlet 3.95 7.65 30.2 0.03 0.08 0.06

- Lines 352-353: Could you provide references to support such hypothesis since you did not measure it?

Answer. We now modified this paragraph to modulate the affirmation.

- Lines 359-361: This sentence should be rewritten in a more prudent way since it is only speculation, you have no clue for the top down control and even though you have PAHs concentrations, the direct link with phytoplankton productivity remains to be demonstrated.

Answer. We now modified the sentence as follow "Finally, it is clear that bottom-up (nutrients and humic substances) drove the differential responses of phytoplankton productivity in the eastern and western part of the lagoon, certainly in conjunction with grazing activity (top-down control").

- Lines 378-380: sounds weird to justify a recent study by an old one... The old reference could be removed without making the sentence meaningless or doubtful.

Answer. We deleted the reference.

- Lines 383-386: These Redfield ratios were not presented in the results sections. They are meaningful and should be presented extensively!

Answer. We added the map for the distribution of this ratio in the lagoon.

- Lines 410-413: That could appear contradictive. If you suppose that higher concentrations of DOC but lower aminopeptidase activity suggest a higher amount of labile organic matter for bacteria, you should clearly state it.

Answer. We now explained that higher DOC concentrations associated with lower aminopeptidase activity suggest a higher amount of labile organic matter for bacteria. The high aminopeptidase activity in the Palizada River plume confirmed the presence of recalcitrant organic matter from terrestrial origin, as opposed to minimum activities in Puerto Real marine waters or in Candelaria mouths, where DOC concentrations were maximal.

- Lines 425-427: So vincinity of Palizada river = phosphatase activity but P-depleted

zone, meaning very low P-availability for phytoplanktonic growth, which seems consistent with le smaller C fixation rates observed in this zone, am I right? If yes, such a link between nutrients and biological productivity could be added to this discussion, this would greatly strengthen the end of the first sub-section which was up to now very speculative. Moreover, this would also strengthen the links between the geochemical and the biological sides of this paper which appear very few connected for the moment.

Answer. Thanks for this pertinent remark that we included in the text.

- End of section 4.3: A small discussion could be added about distance-based effects and community turnover: according to your data, one could think that strong local selective pressures led to strong shift in the community composition, instead of having progressive gradients and/or fast dispersion that would rather just promote a turnover within the active fraction. Such thinking points out the existence of strong driving pressures that you did not measured in your study, since you do not have such strong correlation

Answer. As describe above, we changed the sentence into: "These results indicated that most of the free-living bacterial community detected by molecular fingerprinting (DNA-based) were active (RNA-based) among the lagoon, with the exception of the local transition zones between the lagoon waters and the coastal (El Carmen inlet) or rivers (Palizada and Candelaria)." This sentence points out that some transitions zones strongly affect bacterial diversity and activities. We agree with the reviewer's comment but it is difficult to state on either a strong local selective pressure or progressive gradient driving the strong shift of the community observed in this study. Because our dataset aimed to give an overview of the entire lagoon rather than focusing on the several transition zones of the lagoon, we decided to be cautious and not over interpreting our results.

- Lines 460-466: These two sentences should rather be placed at the beginning of the previous paragraph, before discussing about community structure.

Answer. We moved these sentences at the beginning of the paragraph 4.3 as recommended

- Lines 475-477: Adding a short discussion about "distance-based" similarities in present and active communities, as proposed in a previous comment would perfectly fit with this discussion.

Answer. We added the short discussion suggested

- The link between the severe drought and the potential predictive aspect of this study is not mentioned all along the discussion section, whereas it represents most of your introduction. You need either to remove this "climate-change" part in the intro, or discuss it in the discussion.

Answer. We modified the introduction and discussion in consequence

Conclusion - Lines 512-514: You cannot say that as it was unambiguous, you did not even measure any top-down parameter.

Answer. We modulated our affirmation, in indicating that we supposed a possible shift. In fact we have some signals that confirm this hypothesis; it is the reason why we chose to keep it in the text.

- Lines 514-517: This sentence does not sound correct since you measured C fixation, thus phytoplankton activity. To measure growth, you need repeated measures in time, that you do not have. So you can definitely not state that there is no growth. I understand it is probably not what you wanted to say, but your sentence could be misinterpreted

Answer. We replaced growth by C-fixation

- Lines 528-530: again, you are over confident in your hypothesis: "seems to support", you do not have any temporal data to support this hypothesis.

Answer. done

**TECHNICAL CORRECTIONS**

Answer. All the 15 remarks have been taken into account and fixed

Please also note the supplement to this comment:
http://www.biogeosciences-discuss.net/bg-2016-288/bg-2016-288-AC1-supplement.pdf

―――――――――――――――――

---

## Author Comment (AC2) · 22 Nov 2016

We also printed these comments on a pdf file and added it in supplement (Revewer2.pdf)

SPECIFIC COMMENTS Abstract: - The abstract states that the study will help to "understand how the severe drought period influenced biogeochemical cycling and phyto- and bacterio-plankton communities", nevertheless the study does not compare the data obtained during this period to any other records (maybe due to the fact that they do not exist) obtained during in more "classic" situation. This statement needs to be nuanced.

[Figure]

Answer. Effectively, we have not such data to compare, so we modulated this statement in the text, and added " . . . under such conditions" to be clearer.

- "Coupling between top-down and bottom-up controls accounted for the diverse responses in phytoplankton productivity" : I didn't see any demonstration of top-down control in the study.

Answer. Yes indeed, so we replaced this sentence by "We found that "bottom-up" control accounted for a large part in the variability of phytoplankton productivity". In the discussion, we re written the paragraph 4.1 in order to modulate the top-down control, and considered it as an hypothesis, and not as an affirmation

- The PAH-relative measurements need to be better integrated in the global aim of the study.

Answer. We modified the abstract, introduction and discussion to better integrate our PAH results. In the abstract, we replaced the part "A large set of biogeochemical (nutrients, dissolved and particulate organic matter), phytoplanktonic (biomass and photosynthetic activity) and bacterial (bacterial diversity and ectoenzymatic activities) parameters. . . ." By the part: "A large set of biogeochemical [nutrients, dissolved and particulate organic matter, dissolved polycyclic aromatic hydrocarbons (PAHs)], phytoplanktonic (biomass and photosynthetic activity) and bacterial (bacterial diversity, including PAH-degrading bacteria, and ectoenzymatic activities) parameters. . ." We also added the following sentence: "The highest dissolved total PAH concentrations were measured in El Carmen inlet, suggesting an anthropogenic pollution of the zone probably related to the oil platform exploitation activities in the shallow waters of the South of the Gulf of Mexico."

Introduction - Overall the introduction is too much climate change orientated in regards to the results presented in the manuscript. Introduction needs to be more focused on the local dynamic of this lagoon and on the importance of considering PAH in this specific area. This study is more focused on the dynamic of land/sea transitional coastal

areas under combined climatic and chemical pressures than on climate change effects.

Answer. We deleted the whole paragraph referenced to global change and focused the introduction on the challenge to understand the functioning of continental to marine environments interface in a context of growing anthropization.

- The importance of PAH in the studying area is not mentioned anywhere in the introduction or in the objectives. However, PAH concentrations and PAH-degraders appeared as a substantial part of the results and discussion sections. If PAH is an important factor to consider in the lagoon, this should be clearly presented in the objectives of the study.

Answer. We agree with the Reviewer. We thus added two paragraphs dedicated to PAHs in the introduction: "Local anthropogenic inputs of organic pollutants such as polycyclic aromatic hydrocarbons (PAHs) may also affect bacterial diversity and activities (Lekunberri et al., 2010; Rodríguez-Blanco et al., 2010; Jiménez et al., 2011). Indeed, PAHs, which can comprise as much as 25–35 % of total hydrocarbon content in crude oils (Head et al., 2006), are among the most abundant and ubiquitous pollutants in the coastal environment (González-Gaya et al., 2016). These compounds are recognized by the European and US environmental agencies as priority pollutants for the aquatic medium due to their toxicity, persistence and ability to accumulate in the biota (Kennish, 1992). Hence, the presence of PAHs in the marine environment may induce an increase in the indigenous populations of marine bacteria that can break down and utilize these chemicals as carbon source, the so called "PAH-degrading bacteria" or "PAH degraders". These bacteria are generally strongly selected in oil-impacted ecosystems, where they may account for 70 to 90% of the total bacterial community (Head et al., 2006; Gutierrez et al., 2014)." "Also, Terminos Lagoon is potentially impacted by PAHs, which may come from a diversity of sources including sea-based activities (spills from ships, platforms and pipelines, ballast water discharge, drilling. . .) but also rivers, surface runoffs and the atmosphere that carry various urban and industrial wastes (fuel combustion, traffic exhaust emissions...). Nevertheless, to

our knowledge, little is known about the PAH content in this ecosystem. Even though Noreña-Barroso et al. (1999) have reported PAH concentrations in the American Oysters Crassostrea virginica and Rendon-Von Osten et al. (2007), PAH concentrations in surface sediments, no data are currently available about dissolved PAH concentrations in surface waters of the Terminos Lagoon."

In the objectives, we also added "dissolved PAHs" and "including PAH-degrading bacteria".

We removed one reference from the reference list: Aguayo, P., Gonzalez, C., Barra, R., Becerra, J., and Martinez, M.: Herbicides induce change in metabolic and genetic diversity of bacterial community from a cold oligotrophic lake, World journal of microbiology & biotechnology, 30, 1101-1110, 2014.

And we added several references dealing with PAHs/PAH-degrading bacteria: González-Gaya, B., Fernández-Pinos, M. C., Morales, L., Abad, E., Piña, B., Méjanelle, L., Duarte, C. M., Jiménez, B., and Dachs, J.: High atmosphere-ocean Exchange of semivolatile aromatic hydrocarbons, Nature Geoscience, 9, 438-442, 2016. Gutierrez, T., Rhodes, G., Mishamandani, S., Berry, D., Whitman, W. B., Nichols, P. D., Semple, K. T., and Aitken, M. D.: Polycyclic Aromatic Hydrocarbon Degradation of Phytoplankton-Associated Arenibacter spp. and Description of Arenibacter algicola sp. nov., an Aromatic Hydrocarbon-Degrading Bacterium, Applied and Environmental Microbiology, 80, 618-628, 2014. Head, I. M, Martin Jones, D., and Roling, W. F. M.: Marine microorganisms make a meal of oil, Nature, 4, 173-182, 2006. Jiménez, N., Viñas, M., Guiu-Aragonés, C., Bayona, J. M., Albaigés, J., and Solanas, A. M.: Polyphasic approach for assessing changes in an autochthonous marine bacterial community in the presence of Prestige fuel oil and its biodegradation potential, Applied Microbiolal Biotechnology, 91, 823-834, 2011. Lekunberri, I., Calvo-Díaz, A., Terira, E., Morán, X. A. G., Peters, F., Nieto-Cid, M., Espinoza-González, O., Teixeira, I. G., and Gasol, J.M.: Changes in bacterial activity and community composition caused by exposure to a simulated oil spill in microcosm and mesocosm experiments, Aquatic Microbial Ecology, 59, 169-183, 2010. Noreña-Barroso, E., Gold-Bouchot, G., Zapata-Perez, O., and Sericano, J. L.: Polynuclear Aromatic Hydrocarbons in American Oysters Crassostrea virginica from the Termines Lagoon, Campeche, Mexico, Marine Pollution Bulletin, 38, 637-645, 1999. Rendon-von Osten, J., Memije, M., Ortiz, A., and Benitez, J.: Potential sources of PAHs in sediments from Terminos lagoon, Campeche, Mexico, Toxicology Letters, 172, S162, 2007.

Material and Methods - Section 2.1 (last paragraph): I suggest including the sub-sampling information in the relative subsequent technical sections Is there any available data concerning water circulation in this lagoon ? This information could be relevant for your study

Answer. As suggested, we moved the subsampling informations in each relative subsequent section (Nutrients and POM; see file for details). Concerning the circulation, we added a specific paragraph containing recent reference to describe the sampling zone (section 2.1)= "Recent results on tidal current modeling (Contreras Ruiz Esparza et al., 2014) revealed a dynamic inshore current entering the lagoon through Carmen passage, flowing through the southern half of the lagoon and coming out through Puerto Real and a much slower inverse water current flooding the northern central part of the lagoon. That tidally induced hydrodynamic trend generated a counter clock wise circulation gyre located in the center of the lagoon leeward from Carmen Island."

- Section 2.6: Only free bacteria can be measured according to this protocol not "total bacteria" as mentioned in the text (attached bacteria are not measured)

Answer. Effectively, we agree with reviewer's comment. "Total bacteria" has been changed into "Free-living heterotrophic prokaryote" all through the text to also be in agreement with reviewer 1 remark.

- Section 2.7: In turbid waters, such prefiltration step may lead to the retention of attached bacteria on the filter and induce a bias in bacterial diversity assessment. This method is adequate to collect the bacteria present in aquatic sample but this bias need

to be mentioned in the discussion section.

Answer. As recommended by the reviewer, we pointed this aspect in the discussion section: "In the present study, we focused on the free-living bacteria and disregarded the particle-attached fraction by pre-filtrating the water by 3 $\mu$m, which allowed eliminating the problem of DNA eukaryotic chloroplasts that may have biased our results in the context of gradients of productive zones"

- Section 2.9: The authors used a mixture of 6 PAHs for MPN but there is no explanation of this choice. In addition, what is the ratio of each PAH in the mixture? Why did they use a 10 $\mu$g/mL final concentration (is this concentration realistic when dealing with environmental samples)? Additional information is required in this section.

Answer. This section has been modified as follow: A mixture of 6 PAHs from 2 to 5 rings (naphthalene, fluorene, phenanthrene, fluoranthrene, pyrene and benzo[a]pyrene) prepared in dichloromethane in equimolar concentration was introduced into each well at a final concentration of 10 $\mu$g mL-1, as previously described by Sauret et al. (2016). It corresponds to very high concentration of PAH in nature, i.e. 50 times higher than the values found in the harbour of Leghorn (Cincinelli et al., 2001).

Sauret C, Tedetti M, Guigue C, Dumas C, Lami R, Pujo-Pay M, Conan P, Goutx M, Ghiglione JF (2016) Influence of PAHs among other coastal environmental variables on total and PAH-degrading bacterial communities. Environmental Science and Pollution Research, 23: 4242-4256 Cincinelli, A., Stortini, A.M., Perugini, M., Checcini, L., Lepri, L., 2001. Organic pollutants in sea-surface microlayer and aerosol in the coastal environment of Leghorn (Tyrrhenian Sea). Mar. Chem. 76, 77–98.

Note that all paragraphs referencing to PAH have been changed in the revised version.

SPECIFIC COMMENTS Results - PAH concentrations should be included in section 3.2

Answer. We agree that PAH concentrations could appear in the 3.2 section, but it

is also logical to deal with concentrations and abundance of bacterial PAH-degraders together as we did in 3.5. So, we preferred to not modify.

- Section 3.4: Biomass is the total amount of living material in a given habitat, population, or sample. Specific measures of biomass are expressed in dry weight per unit area of land or unit volume of water. In this section, you report bacterial abundance (cells/ml) and not bacterial biomass.

Answer. It has been fixed in the revised version of the text.

- The standard deviation values for MUF-P and MUF-Lip are higher than the measured mean values. The use of a range of values would be more suitable. In addition the values presented in the text do not fit with the values presented in the legend of Figure 4. This needs to be corrected

Answer. We apologize for the values presented in the legend of Figure 4. We thank the reviewer who realized that these values were wrong and it has now been fixed. Concerning the scale given by means and standard deviations, in fact, we compared the higher values found in Palizada and Chumpan rivers embouchures northward towards El Carmen Island to the rest of the lagoon by giving these mean values. We agree that some of the standard deviation values for MUF-P and MUF-Lip in the rest of the lagoon were higher than the measured mean values, but this is informative of the variability of the data. Giving range of values would not be informative on this aspect.

- Section 3.5: As previously mentioned, flow cytometry does not consider attached bacteria. The protocol used for MPN determination allows the growth of free and attached PAH-degraders. As a consequence, the measured percentage is not accurate.

Answer. As recommended, we modified the whole section and gave precisions concerning the calculation method. A reference has been added in the material and method section, in case the reader wants to have more information. Alexander, M., 1982. Most probable number method for microbial populations. In: Page, A.L., Miller,

R.H., Keeney, D.R. (Eds.), Methods of Soil Analysis, Part 2, 2nd ed. American Society of Agronomy, Madison, WI, pp. 815-820.

Discussion - Lines 352-361: No grazing activity was measured in this study, so this statement need to be presented as a other way to explain the data and not as an affirmation.

Answer. We re-written the paragraph 4.1 in order to modulate the top-down control, and considered it as a hypothesis, and not as an affirmation

- Section 4.3: Please explain the last sentence of the section : " These results indicated that most of the communities detected by molecular fingerprinting were active, with no specific distinction through the lagoon."

Answer. We apologize for being unclear in this sentence, which has been changed to: Here, the combination of DNA and RNA showed similar tendencies within the total and active communities presenting eastern, middle and western distribution among the lagoon. These results indicated that most of the free-living bacterial communities detected by molecular fingerprinting (DNA-based) were active (RNA-based) among the lagoon, with the exception of the local transition zones between the lagoon waters and the coastal (El Carmen inlet) or rivers (Palizada and Candelaria).

Conclusion - No demonstration of any top-down control in this study

Answer. It has been modified everywhere in the text as previously detailed

- The conclusion gives a nice overview of the main results obtained in the lagoon but does not respond to the climate change questions mentioned in the introduction. This needs clarification or rephrasing of the introduction.

Answer. It has been modified as suggested.

TECHNICAL AND TYPOS CORRECTIONS

Answer. All the 9 remarks have been taken into account and fixed

Please also note the supplement to this comment:
http://www.biogeosciences-discuss.net/bg-2016-288/bg-2016-288-AC2-
supplement.pdf
* * *

---

## Author Comment (AC3) · 22 Nov 2016

Title:

5

Biogeochemical cycling and phyto- and bacterio-plankton communities in a large and shallow tropical lagoon (Terminos Lagoon, Mexico) under 2009-2010 El Niño Modoki drought conditions

**Authors:**

Pascal Conan1, Mireille Pujo-Pay1, Marina Agab1, Laura Calva-Benítez2, Sandrine Chifflet3,

10 Pascal Douillet3, Claire Dussud1, Renaud Fichez3, Christian Grenz3, Francisco Gutierrez Mendieta2, Montserrat Origel-Moreno2,3, Arturo Rodríguez-Blanco1, Caroline Sauret1, Tatiana Severin1, Marc Tedetti3, Rocío Torres Alvarado2, Jean-François Ghiglione1

[revised manuscript text omitted]
 is done by the use of sybr green which induces a green fluorescence and then prokaryotes are separated using the SSC diffraction parameter.

**2.7 Total and metabolically active bacterial community structure**

Nucleic acids were extracted on 0.2 µm-pore-size filter (47 mm, PC, Nucleopore) by filtration of 1 L of pre-filtered (3 µm) water. Co-extraction of DNA and RNA was performed after chemical cell lysis (Ghiglione et al., 1999) with the Qiagen Allprep DNA/RNA extraction kit using manufacturer instructions. DNA and cDNA (by M-MLV reverse transcription of 16S rRNA, Promega) were used as a template for PCR amplification of the variable V3 region of the 16S rRNA gene (*Escherichia coli* gene positions 329–533; Brosius et al., 1981). The primer w34 was fluorescently labelled at the 5'-end position with phosphoramidite (TET, Applied Biosystems). CE-SSCP analysis was performed using the 310 Genetic Analyzer and Genescan analysis software (Applied Biosystems), as previously described (Ortega-Retuerta et al., 2012).

**2.8 Extracellular enzymatic activities**

220 Aminopeptidase,  $\beta$ -glucosidase and lipase were measured using a VICTOR3 spectrofluorometer (Perkin Elmer) after incubations of 2 h at *in situ* temperature with L-leucine-7amido-4-methyl coumarin (LL, 5  $\mu$ M final), MUF- $\beta$ -D-glucoside ( $\beta$ -Glc, 0.25  $\mu$ M final) or MUF-palmitate (Lip, 0.25  $\mu$ M final). These saturated concentrations and optimized time incubations were determined prior to the extracellular enzymatic activities measurement, as previously described (Van Wambeke et al., 2009).

225

**2.9 Quantification of PAH-degrading bacteria by Most-Probable-Number**

The quantification of PAH-degrading bacteria was performed by the most-probable-number (MPN). A total of 100 µL of each sample was introduced in triplicate in a 48-well microplate with 900 µL of sterile minimum medium, as previously described (Rodríguez-Blanco et al., 2010; Sauret et al., 2016). A mixture of 6
PAHs from 2 to 5 rings (naphthalene, fluorene, phenanthrene, fluoranthrene, pyrene and benzo[a]pyrene) prepared in dichloromethane in equimolar concentration was introduced into each well at a final concentration of 10 µg mL-1, as previously described by Sauret et al. (2016). It corresponds to very high concentration of PAH in nature, i.e. 50 times higher than the values found in the harbour of Leghorn (Cincinelli et al., 2001). After 2 weeks of incubation, the change from blue to pink, indicating oxidation of the resazurin contained in the medium
was checked and each sample was analysed by flow cytometry. Classical MPN table gave the most probable number of bacteria able to degrade the mixture of six PAHs (Alexander, 1982).

**2.10 Statistical analysis**

- Comparative analysis of 16S rDNA- or 16S rRNA-based CE-SSCP fingerprints was carried out with the 240 PRIMER 6 software (PRIMER-E, Ltd., UK) using Bray-Curtis similarities. We used the similarity profile test SIMPROF (PRIMER 6) to test the null hypothesis of randomly that a specific sub-cluster can be recreated by permuting the entry ribotypes and samples, when using hierarchical agglomerative clustering. The significant branch (SIMPROF, p<0.05) was used as a prerequisite for defining bacterial clusters, and clusters were reported on non-metric multidimensional scaling (MDS) representation.
- 245 Canonical correspondence analysis (CCA) was used to investigate the variations in the CE-SSCP profiles under the constraint of our set of environmental variables, using CANOCCO software (version 5.0), as previously described in Berjeb et al. (2011). Significant variables (i.e. variables that significantly explained changes in 16S rDNA- and 16S rRNA-based fingerprintings) in our data set were chosen using a forward-selection procedure. Explanatory variables were added until further addition of variables failed to contribute significantly (p< 0.05) to a substantial improvement to the model's explanatory power. Environmental parameters were previously transformed according to their pairwise distributions, and Spearman's rank pairwise correlations between the transformed environmental variables were used to determine their significance with

**255**

260

**3. Results**

Statel v2.7.

**3.1 Distribution of physical parameters**

At the studied period, Terminos Lagoon was characterized by a North West-South East positive gradient of temperature from >30 to about 27°C (Fig. 2A). Salinity was maximal at Puerto Real inlet (37.50) and along the southern limits of El Carmen Island, intermediate at Candelaria and Chumpan River mouths, and minimal (21.57) close to the Palizada River (Fig. 2B).

**3.2 Distribution of biogeochemical parameters**

Nitrate and ammonium concentrations (Fig. 2C and 2D) were maximum close to the Palizada embouchure (16.6 and 0.3  $\mu$ M, respectively) and to the Puerto Real inlet (2.5  $\mu$ M in NO3 and the highest NH4 concentration of 1  $\mu$ M). In the rest of the lagoon, NO3 
[revised manuscript text omitted]

---

## Author Response (AR2)

**Associate Editor Decision: Publish subject to minor revisions (Editor review)**
(15 Dec 2016) by Dr Clare Woulds  // Comments to the Author:

Dear Dr Conan

Thank you for submitting your revised manuscript and for the changes that you have made in response to reviewer comments. I feel the manuscript is almost ready, but there are several points where further clarification or alteration are still required. I would therefore like to invite you to submit a further version in which you have attended to my comments listed below.

I suggest a re-phrasing of the opening sentence of the abstract, the meaning is still not clear.
The first sentence has been modified

Lines 683-685. This new statement needs proof reading for use of English, as it currently may not mean much to readers who have not read the reviews and your responses to them.
This paragraph has been modified in order to be comprehensive for new readers

Section 2.9. The wording here still suggests that flow cytometry was conducted after 2 weeks of incubation with the PAH mixture. If this is not in fact the case (as suggested in your response) please alter the wording here to make it very clear when the flow cytometry was conducted.
You are right. As usually proceed in this method, the abundance of bacteria (by flow cytometry) was performed at the beginning of the experiment only. The results were based on the visual observation of color changes after 2 weeks of incubation in the serial dilution and on the application of the traditional MPN table, as previously described in a recent paper published by our team (Sauret et al. 2016). The description of the method has been improved to make this point clearer.

Line 884. Why are DOC and DON concentrations given as percentages here? Please check and correct units if necessary, or add explanation for the use of percentages.
Yes, it was a mistake remaining from the previous version where we compared the concentrations of the two areas. The units are now corrected

Line 925. I think this should say 'and to a greater extent', as the correlations with phosphatatse are stronger than the ones with aminopeptidase.
It has been modified

Line 1006. Please change the word 'certainly' for 'possibly' or 'probably'.
It has been modified

I am not sure what point is made by the N sink section ending line 1035. It doesn't seem to directly concern data presented in this manuscript, and does not seem to reach a tangible conclusion. Please clarify the importance of this section or consider removing it.
We agree with your remark. After careful read, we have chosen to delete this sentence as suggested.

Line 1085. It is not clear which zone you are referring to here. Is the 'zone' referred to here the area around the Palizada, or that around the Puerto Real inlet? How do you explain / what is the difference between these zones in terms of P sources? If your text already answers this query then please clarify it.

This paragraph has been modified to precise our idea concerning the difference in the limitation of the two zones

It is still not clear to me how you can assert that Terminos Lagoon was a C sink and a N-assimilator, as you have not measured the sink fluxes for either C or N. These terms have been removed from other parts of the manuscript but remain in the conclusions section. Please explain how these conclusions have been reached.

The term C-sink was not indicated in the conclusion in the previous version, only the term "sink" appeared. In order to avoid confusion, we have changed the sentence by
"Hence during our study, the water column of Terminos Lagoon functioned globally as a kind of "nitrogen assimilator"".

If possible, please have the manuscript checked for use of English. As some copy editing is required.

As we had a delay to submit our revised version, we paid an official translator to get an improvement on our manuscript. Corrections are marked in the hereafter files with tracking changes

**Title:**

[revised manuscript text omitted]

**2.2 Nutrients and Dissolved organic matter**

As soon as the sampling Niskin sampler was retrieved on board, a previously acid washed 40 mL Schott® glass vial previously acid washed was rinsed with sampled water, filled, immediately injected with the fluorometric detection reagent for ammonia determination (as described in Holmes et al., 1999), sealed, and stored in the dark for later analysis at in the laboratory. ThenFollowing this, two 30 mL and one 150 mL plastic acid washed vials were then rinsed with sampled water, filled, stored in a specifically dedicated and refrigerated ice cooler, to be later deep-frozen at in the laboratory while awaiting analysis of dissolved inorganic and organic nutrients, as follows:.

**Commentaire [MR2]:** You should standardise the form of this to be like line 180 by moving the next paragraph (lines 167-171) up to this line

Nitrate ($NO_3 \pm 0.02$ µM), nitrite ($NO_2 \pm 0.01$ µM), phosphate ($PO_4 \pm 0.01$ µM) and silicate ($Si(OH)_4 \pm 0.05$ µ
[revised manuscript text omitted]

**Commentaire [MR13]:** Do you mean 'to the southwest of CI' ? ie, offshore CI (rather than on the island, as you suggest)?

intermediated communities found between  El Carmen Inlet and the Palizada River in the western part of the

360 lagoon (cluster V; stations 2, 4, 6) and in the middle of the lagoon, close to the Candelaria River (cluster IV; stations 22, 24, 27).

Metabolically active bacterial communities as a function of 16S rRNA-based fingerprints singled out 2 stations (Palizada River and El Carmen Inlet) and aggregated 5 groups of stations which are slightly different from the DNA-based clusters (Fig. 6B). Three of thse groups included a large number of samples: cluster I

365 formed the largest cluster with 15 stations located in the Eastern part of the lagoon; cluster II grouped 9 stations in the middle of the lagoon North  of Chumpan river to  Carmen Island; cluster III grouped 5 stations in the North Western part of the lagoon, close to El Carmen inlet. Two other groups with fewer stations showed intermediate communities found close to the Palizada river mouth (cluster IV; stations 6 and 8) and further east (cluster V; stations 9 and 12).

370

**3.7 Environmental drivers of the total and active prokaryote community structures**

To analyse the main environmental factors controlling the spatial distribution of total (Fig. 7A) and active (Fig. 7B) prokaryote communities, we performed a canonical correspondence analysis (CCA). In both DNA- and RNA- based analysis, the cumulative percentage of variance of the species-environment relationship

375 indicated that the first and second canonical axis explained 48 % and 24 % of the total variance, respectively for DNA and 45 % and 31 % for RNA. The remaining axes accounted for less than 14 % of the total variance each, and thus were not considered as significant enough.

In the DNA-based CCA, the first canonical axis was positively correlated with $NO_3^-$ and CHL and negatively correlated with concentration of DOC, DOP, DON and oxygen. In the RNA-based CCA, the first

380 canonical axis was positively correlated with $NO_3^-$ and PAHs and negatively correlated with the concentration of POC, PON, oxygen, salinity, $PO_4$ and CHL. The concomitant effect of those parameters explained 27 % and 40 % (ratio between the sum of all canonical eigenvalues and the sum of all eigenvalues) of the changes in bacterial community structure found in the DNA- and RNA-based fractions, respectively (Figure 7).

385

**4. Discussion**

**4.1 Biogeochemical characteristics of Terminos Lagoon under low river discharge conditions**

With a contribution of about 76 % to river inputs in the lagoon (Fichez et al., 2016; Jensen et al., 1989),
Palizada River delivers most of the new nitrogen inputs as nitrate and ammonium. High concentrations in

390 nitrogen were also measured in the Puerto Real Iinlet, suggesting a second nitrogen source from coastal seawater. These two sources have clearly different impacts on primary producer development and activity as shown by the Phaeo:CHL ratio (<20 % in the vicinity of the Palizada River, but >30 % close to the Puerto Real Iinlet) and $P^b_m$ values (low in the Palizada area and higher close to the inlet). So, despite greater chlorophyll degradation (indicated by high Phaeo concentrations), phytoplanktonic cells were more productive under the

395 influence of waters from the Gulf of Mexico when compared to those under the river's influence. Specifically, there were similar nutrients, DOM and POM concentrations for the two zones and we measured a similar potential primary production per unit volume (27.5 and 30.2 mgC m$^{-3}$ h$^{-1}$ for Palizada River and Puerto Real Inlet, respectively). However, the chlorophyll stock was about 2-fold lower in the area of the inlet (6.3 and 3.9 mgCHL m$^{-3}$ for Palizada River and Puerto Real Inlet, respectively). Specifically, in terms of average values for

400 the two zones, with similar nutrients, DOM and POM concentrations, we measured a similar potential primary production per unit volume (27.5 and 30.2 mgC m$^{-3}$ h$^{-1}$ for Palizada River and Puerto Real Iinlet, respectively),

**Commentaire [MR14]:** Axes ?

**Commentaire [MR15]:** This is awkward and you might want to reword it. I'm not sure to understand well enough to attempt to do so myself. There are too many commas…

[revised manuscript text omitted]

**Commentaire [MR19]:** per cell ?

**Commentaire [MR20]:** the Candelaria R has more than one mouth ? if so, consider saying ' in the mouths of the CR'

Phosphatase activity is well known to be controlled by the availability of soluble reactive phosphorus (Van Wambeke et al., 2009). This activity was essentially observed in the vicinity of the Palizada River, which is the main source of PP in the lagoon, and but not in Puerto Rreal Iinlet, the two $PO_4$-depleted zones which indirectly influence the stoichiometry of particulate organic matter, as discussed above (Fig. 8). Thus, a zone with clear phosphatase activity but which is P-depleted means very low P-availability for phytoplanktonic growth. Thus, a clear phosphatase activity but P-depleted zone means very low P-availability for phytoplanktonic growth. This observation is consistent with the low phytoplankton productivity observed, indicating a weak C-fixation rates in this the zone Palizada mouth, which strengthens our bottom-up control hypothesis. Extracellular phosphatase activity was significantly ($p<0.05$, $n=35$) negatively correlated with $PO_4$ ($\rho=-0.46$) and positively correlated with PP ($\rho=0.60$). Our data therefore converge with the model previously proposed by Robadue *et al.* (2004) predicting which predicted a different behaviour between the eEastern and wWestern sides of the lagoon in terms of both water budget as well as and ecosystem functioning, this a distinction being which is mostly driven by the respective influences of the Palizada River discharge in the wWest and the Puerto Real marine water inputs in the nNorth eEast.

Commentaire [MR21]: Something seems to be missing here. Do you mean "which are the two…"
Or
…Inlet. These are the two…

[revised manuscript text omitted]

Figure 2

[Figure]

Figure 3

[Figure]

Figure 4

[Figure]

945

Figure 5

[Figure]

950

Figure 6

[Figure]

955

Figure 7

[Figure]

Figure 8

[Figure]